# Fractal Structure and Generalization Properties of Stochastic Optimization Algorithms

**Alexander Camuto**[1], **George Deligiannidis**[1], **Murat A. Erdogdu**[2],
**Mert Gürbüzbalaban**[3✉], **Umut Şimşekli**[4✉], **Lingjiong Zhu**[5]

**1:** University of Oxford & Alan Turing Institute   **2:** University of Toronto & Vector Institute
**3:** Rutgers Business School   **4:** INRIA & École Normale Supérieure - PSL Research University
**5:** Florida State University

The authors are in alphabetical order.
✉: Corresponding authors

## Abstract

Understanding generalization in deep learning has been one of the major challenges in statistical learning theory over the last decade. While recent work has illustrated that the dataset and the training algorithm must be taken into account in order to obtain meaningful generalization bounds, it is still theoretically not clear which properties of the data and the algorithm determine the generalization performance. In this study, we approach this problem from a dynamical systems theory perspective and represent stochastic optimization algorithms as *random iterated function systems* (IFS). Well studied in the dynamical systems literature, under mild assumptions, such IFSs can be shown to be ergodic with an invariant measure that is often supported on sets with a *fractal structure*. As our main contribution, we prove that the generalization error of a stochastic optimization algorithm can be bounded based on the 'complexity' of the fractal structure that underlies its invariant measure. Then, by leveraging results from dynamical systems theory, we show that the generalization error can be explicitly linked to the choice of the algorithm (e.g., stochastic gradient descent – SGD), algorithm hyperparameters (e.g., step-size, batch-size), and the geometry of the problem (e.g., Hessian of the loss). We further specialize our results to specific problems (e.g., linear/logistic regression, one hidden-layered neural networks) and algorithms (e.g., SGD and preconditioned variants), and obtain analytical estimates for our bound. For modern neural networks, we develop an efficient algorithm to compute the developed bound and support our theory with various experiments on neural networks.

## 1 Introduction

In statistical learning, many problems can be naturally formulated as a risk minimization problem

$$\min_{w \in \mathbb{R}^d} \Big\{ \mathcal{R}(w) := \mathbb{E}_{z \sim \pi}[\ell(w, z)] \Big\}, \tag{1}$$

where $z \in \mathcal{Z}$ denotes a data sample coming from an unknown distribution $\pi$, and $\ell : \mathbb{R}^d \times \mathcal{Z} \to \mathbb{R}_+$ is the composition of a loss and a function from the hypothesis class parameterized by $w \in \mathbb{R}^d$. Since the distribution $\pi$ is unknown, one needs to rely on empirical risk minimization as a surrogate to (1),

$$\min_{w \in \mathbb{R}^d} \Big\{ \hat{\mathcal{R}}(w, \mathbf{S}_n) := (1/n) \sum_{i=1}^n \ell(w, z_i) \Big\}, \tag{2}$$

where $\mathbf{S}_n := \{z_1, \ldots, z_n\}$ denotes a *training set* of $n$ points that are independently and identically distributed (i.i.d.) and sampled from $\pi$, and model training often amounts to using an optimization algorithm to solve the above problem.

35th Conference on Neural Information Processing Systems (NeurIPS 2021).

Statistical learning theory is mainly interested in understanding the behavior of the *generalization error*, i.e., $\hat{\mathcal{R}}(w, \mathbf{S}_n) - \mathcal{R}(w)$. While classical results suggest that models with large number of parameters should suffer from poor generalization [SSBD14, AB09], modern neural networks challenge this classical wisdom: they can fit the training data perfectly, yet manage to generalize well [ZBH+17, NBMS17]. Considering that the generalization error is influenced by many factors involved in the training process, the conventional algorithm- and data-agnostic uniform bounds are typically overly pessimistic in a deep learning setting. In order to obtain meaningful, non-vacuous bounds, the underlying data distribution and the choice of the optimization algorithm need to be incorporated in the generalization bounds [ZBH+17, DR17].

Our goal in this study is to develop novel generalization bounds that explicitly incorporate the data and the optimization dynamics, through the lens of dynamical systems theory. To motivate our approach, let us consider stochastic gradient descent (SGD), which has been one of the most popular optimization algorithms for training neural networks. It is defined by the following recursion:

$$w_k = w_{k-1} - \eta \nabla \tilde{\mathcal{R}}_k(w_{k-1}), \quad \text{where} \quad \nabla \tilde{\mathcal{R}}_k(w) := \nabla \tilde{\mathcal{R}}_{\Omega_k}(w) := (1/b) \sum_{i \in \Omega_k} \nabla \ell(w, z_i).$$

Here, $k$ represents the iteration counter, $\eta > 0$ is the step-size (also called the learning-rate), $\nabla \tilde{\mathcal{R}}_k$ is the stochastic gradient, $b$ is the batch-size, and $\Omega_k \subset \{1, 2, \ldots, n\}$ is a random subset drawn with or without replacement with cardinality $|\Omega_k| = b$ for all $k$.

Constant step-size SGD forms a Markov chain with a stationary distribution $w_\infty \sim \mu$, which exists and is unique under mild conditions [DDB20, YBVE20], and intuitively we can expect that the generalization performance of the trained model to be intimately related to the behavior of the risk $\mathcal{R}(w)$ over this limit distribution $\mu$. In particular, the Markov chain defined by the SGD recursion can be written by using random functions $h_{\Omega_k}$ at each SGD iteration $k$, i.e.,

$$w_k = h_{\Omega_k}(w_{k-1}), \quad \text{with} \quad h_{\Omega_k}(w) = w - \eta \nabla \tilde{\mathcal{R}}_k(w). \tag{3}$$

Here, the randomness in $h_{\Omega_k}$ is due to the selection of the subset $\Omega_k$. In fact, such a formulation is not specific to SGD; it can cover many other stochastic optimization algorithms if the random function $h_{\Omega_k}$ is selected accordingly, including second-order algorithms such as *preconditioned SGD* [Li17]. Such random recursions (3) and characteristics of their stationary distribution have been studied extensively under the names of *iterated random functions* [DF99] and *iterated function system* (IFS) [Fal04]. In this paper, from a high level, we relate the 'complexity of the stationary distribution' of a particular IFS to the generalization performance of the trained model.

We illustrate our context in two toy examples. In the first one, we consider a 1-dimensional quadratic problem with $n = 2$ and $\ell(w, z_1) = w^2/2$ and $\ell(w, z_2) = w^2/2 - w$. We run SGD with constant step-size $\eta$ to minimize the resulting empirical risk. We simply choose $\Omega_k \subset \{1, 2\}$ uniformly random with batch-size $b = 1$, and we plot the histograms of stationary distributions for different step-size choices $\eta \in \{1/100, 1/3, 2/3\}$ in Figure 1. We observe that the *support* of the stationary distribution of SGD depends on the step-size: As the step-size increases the support becomes less dense and a *fractal structure* in the stationary distribution can be clearly observed.

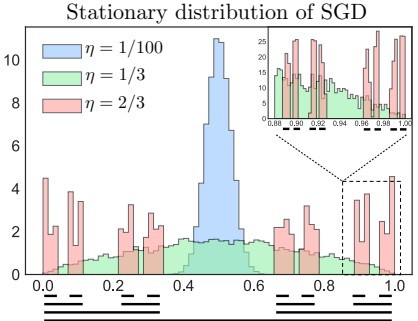

Figure 1: Middle-third Cantor set as the support of the stationary distribution of constant step-size SGD for $\ell(w, z_1) = w^2/2$ and $\ell(w, z_2) = w^2/2 - w$.

This behavior is not surprising, at least for this toy example. It is well-known that the set of points that is invariant under the resulting IFS (termed as the attractor of the IFS) for the specific choice of $\eta = 2/3$ is the famous 'middle-third Cantor set' [FW09], which coincides with the support of the stationary distribution of the SGD.

As another example, we run SGD with constant step-size in order to train an ordinary linear regression model for a dataset of $n = 5$ samples and $d = 2$ dimensions, i.e., $a_i^\top w \approx y_i$, where for $i = 1, \ldots, 5$, $y_i$ and each coordinate of $a_i$ are drawn uniformly at random from the interval $[-1, 1]$. Figure 2 shows the heatmap of the resulting stationary distributions for different step-size choices $\eta$ ranging from 0.1 to 0.9 (bright colors represent higher density). We observe that for small step-size choices, the stationary distribution is dense, whereas a fractal structure can be clearly observed as the step-size gets larger.

Fractals are complex patterns and the level of this complexity is typically measured by the *Hausdorff dimension* of the fractal, which is a notion of dimension that can take fractional values[1], and can be much smaller than the ambient dimension $d$. Recently, assuming that SGD trajectories can be well-approximated by a certain type of stochastic differential equations (SDE) [ŞGN+19, ŞSG19, NcGR19, ŞZTG20], it is shown that the generalization error can be controlled by the Hausdorff dimension of the trajectories of the SDE, instead of their ambient dimension $d$ [ŞSDE20]. That is, the ambient dimension that appears in classical learning theory bounds is replaced with the Hausdorff dimension.

The fractal geometric approach presented in [ŞSDE20] can capture the 'low dimensional structure' of fractal sets and provides an alternative perspective to the compression-based approaches that aim to understand why overparametrized networks do not overfit [AGNZ18, SAN20, SAM+20, HJTW21]. However, SDE approximations for SGD often serve as mere heuristics, and guaranteeing a good approximation typically requires unrealistically small step-sizes [LTE19]. For more realistic step-

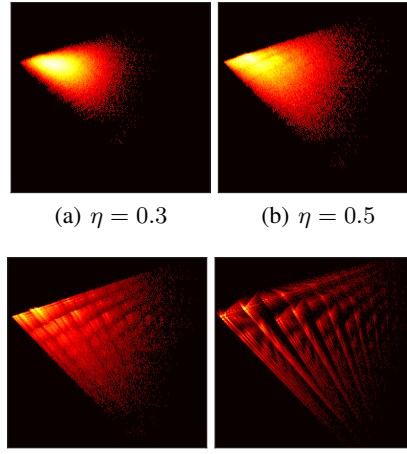

(a) $\eta = 0.3$  (b) $\eta = 0.5$

(c) $\eta = 0.7$  (d) $\eta = 0.9$

Figure 2: The stationary distribution of constant step-size SGD for linear regression, where we have $n = 5$ data points and $w \in \mathbb{R}^2$.

sizes, theoretical concerns have been raised about the validity of conventional SDE approximations for SGD [LMA21, GŞZ21, Yai19]. Another drawback of the SDE approximation is that the bounds in [ŞSDE20] are implicit, in the sense that they cannot be related to algorithm hyperparameters, problem geometry, or data.

We address these issues and present a direct, *discrete-time* analysis by exploiting the connections between IFSs and stochastic optimization algorithms. Our contributions are summarized as follows:

- We extend [ŞSDE20] and show that the generalization error can be linked to the Hausdorff dimension of *invariant measures* (rather than the Hausdorff dimension of *sets* as in [ŞSDE20]). More precisely, under appropriate conditions, we establish a generalization bound for the stationary distribution of IFS $w_\infty \sim \mu$. That is, with probability at least $1 - 2\zeta$,

$$|\hat{\mathcal{R}}(w_\infty, \mathbf{S}_n) - \mathcal{R}(w_\infty)| \lesssim \sqrt{\frac{\overline{\dim}_{\mathrm{H}}\mu \, \log^2(n)}{n} + \frac{\log(1/\zeta)}{n}}, \tag{4}$$

for $n$ large enough, where $\overline{\dim}_{\mathrm{H}}\mu$ is the (upper) Hausdorff dimension of the measure $\mu$.

- By leveraging results from IFS theory, we further link $\overline{\dim}_{\mathrm{H}}\mu$ to (i) the form of the recursion (e.g., $h_{\Omega_k}$ in (3)), (ii) algorithm hyperparameters (e.g., $\eta$, $b$), and (iii) problem geometry (e.g., Hessian of $\tilde{\mathcal{R}}_k$), through a single term, which encapsulates all these components and their interaction.

- We establish bounds on $\overline{\dim}_{\mathrm{H}}\mu$ for SGD and preconditioned SGD algorithms, when used to minimize various empirical risk minimization problems such as least squares, logistic regression, support vector machines. In all cases, we explicitly link the generalization performance of the model to the hyperparameters of the underlying training algorithm.

- Finally, we numerically compute key quantities that appear in our generalization bounds, and show empirically that they have a statistically significant correlation with the generalization error.

**Notation and preliminaries.** $B_d(x, r) \subset \mathbb{R}^d$ denotes the closed ball centered around $x \in \mathbb{R}^d$ with radius $r$. A function $f : \mathbb{R}^{d_1} \to \mathbb{R}^{d_2}$ is said to be (Fréchet) differentiable at $x \in \mathbb{R}^{d_1}$ if there exists a $d_1 \times d_2$ matrix $J_f(x) : \mathbb{R}^{d_1} \to \mathbb{R}^{d_2}$ such that $\lim_{\|h\| \to 0} \|f(x + h) - f(x) - J_f(x)h\| / \|h\| = 0$. The matrix $J_f(x)$ is called the differential of $f$, also known as the Jacobian matrix at $x$, and determinant of $J_f(x)$ is called the Jacobian determinant [HS74]. For real-valued functions $f, g$, we define $f(n) = \omega(g(n))$ if $\lim_{n \to \infty} |f(n)|/g(n) = \infty$. For a set $A$, $|A|$ denotes its cardinality. For a scalar-valued function $\tilde{f} : \mathbb{R} \to \mathbb{R}$, we define $\|\tilde{f}\|_\infty = \max_{r \in \mathbb{R}} |\tilde{f}(r)|$.

---

[1]The Hausdorff dimension of the middle-third Cantor set in Figure 1 is $\log_3(2) \approx 0.63$ whereas the ambient dimension is 1 [Fal04, Example 2.3].

## 2 Technical Background on Fractal Geometry

Fractal sets emerge virtually in all branches of science, and fractal-based techniques have been used in machine learning [SHTY13, MSS19, DSD20, ŞSDE20, AGZ21]. The inherent 'complexity' of a fractal set often plays an important role and it is typically measured by its *fractal dimension*, where several notions of dimension have been proposed [Fal04]. In this section, we briefly mention two important notions of fractal dimension, which will be used in our theoretical development.

**Minkowski dimension of a set.** The Minkowski dimension (also known as the box-counting dimension [Fal04]) is defined as follows. Let $F \subset \mathbb{R}^d$ be a set and for $\delta > 0$, let $N_\delta(F)$ denote a collection of sets that contains the smallest number of closed balls of diameter at most $\delta$ which cover $F$. Then the upper-Minkowski dimension of $F$ is defined as follows:

$$\overline{\dim}_{\mathrm{M}}F := \limsup_{\delta \to 0}\Big[\log|N_\delta(F)| \,/\, \log(1/\delta)\Big]. \tag{5}$$

To visualize the upper-Minkowski dimension of a set $F$, consider the set $F$ lying on an evenly spaced grid and count how many boxes are required to cover the set. The upper-Minkowski dimension measures how this number changes as the grid is made finer using a box-counting algorithm.

**Hausdorff dimension of a set.** An alternative to the purely geometric Minkowski dimension, the Hausdorff dimension [Hau18] is a measure theoretical notion of fractal dimension. It is based on the *Hausdorff measure*, which generalizes the traditional notions of area and volume to non-integer dimensions [Rog98]. More precisely, for $s \geq 0$, let $F \subset \mathbb{R}^d$ and $\delta > 0$, and denote $\mathcal{H}^s_\delta(F) := \inf \sum_{i=1}^\infty \mathrm{diam}(A_i)^s$, where the infimum is taken over all the $\delta$-coverings $\{A_i\}_i$ of $F$, that is, $F \subset \cup_i A_i$ with $\mathrm{diam}(A_i) < \delta$ for every $i$. The $s$-dimensional Hausdorff measure of $F$ is defined as the monotone limit $\mathcal{H}^s(F) := \lim_{\delta \to 0} \mathcal{H}^s_\delta(F)$. When $s \in \mathbb{N}$, $\mathcal{H}^s$ is the $s$-dimensional Lebesgue measure up to a constant factor; hence the generalization of 'volume' to fractional orders.

Based on the Hausdorff measure, the *Hausdorff dimension* of a set $F \subset \mathbb{R}^d$ is then defined as follows:

$$\dim_{\mathrm{H}} F := \sup\{s > 0 : \mathcal{H}^s(F) > 0\} = \inf\{s > 0 : \mathcal{H}^s(F) < \infty\}.$$

In other words, the Hausdorff dimension of $F$ is the 'moment' $s$ when $\mathcal{H}^s(F)$ drops from $\infty$ to $0$, that is, $\mathcal{H}^r(F) = 0$ for all $r > \dim_{\mathrm{H}} F$ and $\mathcal{H}^r(F) = \infty$ for all $r < \dim_{\mathrm{H}} F$.

We always have $0 \leq \dim_{\mathrm{H}} F \leq d$, and when $F$ is bounded, we always have $0 \leq \dim_{\mathrm{H}} F \leq \overline{\dim}_{\mathrm{M}}F \leq d$ [Fal04]. Furthermore, the Hausdorff dimension of $\mathbb{R}^d$ equals $d$, and the Hausdorff dimension of smooth Riemannian manifolds correspond to their intrinsic dimension, e.g. $\dim_{\mathrm{H}} \mathbb{S}^{d-1} = d - 1$, where $\mathbb{S}^{d-1}$ is the unit sphere in $\mathbb{R}^d$.

**Hausdorff dimension of a probability measure.** IFSs generate invariant measures as the number of iterates goes to infinity, and random fractals arise from such invariant measures. There has been a growing literature that studies the structure of such random fractals [Saz00, NSB02, MS02, Ram06, FST06, JR08], where the notion of fractal dimension can be extended to measures, and our theory will rely on the Hausdorff dimension of invariant measures associated with stochastic optimization algorithms. In particular, we will mainly use the *upper Hausdorff dimension* $\overline{\dim}_{\mathrm{H}}\mu$ of a Borel probability measure $\mu$ on $\mathbb{R}^d$, which is defined as follows: $\overline{\dim}_{\mathrm{H}}\mu := \inf\{\dim_{\mathrm{H}} A : \mu(A) = 1\}$. In other words, $\overline{\dim}_{\mathrm{H}}\mu$ is the smallest Hausdorff dimension of all measurable sets with full measure.

## 3 Generalization Bounds for Stochastic Optimization Algorithms as IFSs

In this section, we will present our main theoretical results which relate the generalization error to the upper-Hausdorff dimension of the invariant measure associated with a stochastic optimization algorithm. We consider a standard supervised learning setting, where $\mathcal{Z} = \mathcal{X} \times \mathcal{Y}$, where $\mathcal{X}$ is the space of features and $\mathcal{Y}$ is the space of labels, and $\pi$ is the unknown data distribution on $\mathcal{Z}$.

For mathematical convenience, in order to construct the training set with $n$ elements, we first consider an *infinite sequence* of i.i.d. data samples from the data distribution $\pi$, then we take the first $n$ elements from this infinite sequence. More precisely, we consider the (countable) product measure $\pi^\infty = (\pi \otimes \pi \otimes \dots)$ defined on the cylindrical sigma-algebra. Then, we consider the infinite i.i.d. data sequence as $\mathbf{S} \sim \pi^\infty$, i.e., $\mathbf{S} = (z_j)_{j \geq 1}$ with $z_j \overset{\mathrm{i.i.d.}}{\sim} \pi$ for all $j = 1, 2, \dots$. Finally, we define the training set as $\mathbf{S}_n := (z_1, \dots, z_n)$, i.e., we take the first $n$ elements of $\mathbf{S}$. To avoid technical

complications, throughout the paper we will assume that all the encountered functions and sets are measurable. All the proofs are given in the supplement.

Given a dataset $\mathbf{S}_n$, we represent the *training algorithm* as an IFS, which is based on the following recursion: $w_k = h_{\Omega_k}(w_{k-1}; \mathbf{S}_n)$, where the mini-batch $\Omega_k$ is i.i.d. sampled according to some distribution (e.g., sampling without-replacement uniformly among all possible mini-batches). This compact representation enables us to cover a broad range of optimization algorithms with a unified notation, including SGD (see (3)), as well as preconditioned SGD, and stochastic Newton methods. For example, if we take $h_{\Omega_k}(w; \mathbf{S}_n) = w - \eta H_k(w)^{-1} \nabla \tilde{\mathcal{R}}_k(w)$ where $H_k(w)$ is an estimate of the Hessian of $\tilde{\mathcal{R}}_k$, we cover stochastic Newton methods [EM15]. Similar constructions can be made for other popular algorithms, such as SGD-momentum [Qia99], RMSProp [HSS12], or Adam [KB15].

Notice that there are only finitely many values that $\Omega_k$ can take. For example, in the case of without-replacement mini-batch sampling with batch-size $b$, there are in total $m_b = m_b^{\text{wo-replacement}} := \binom{n}{b}$ many subsets of $\{1, 2, \ldots, n\}$ with cardinality $b$. Alternatively, another setup would be to divide the dataset into $m_b = m_b^{\text{batch}} := n/b$ batches with each batch having $b$ elements, and at each iteration $k$, we can randomly choose one of the batches. In both examples we can enumerate as $S_1, S_2, \ldots, S_{m_b}$. If the probability of sampling the mini-batch $\Omega_k = S_i$ is $p_i$ for every $i$, then, with a slight abuse of notation, we can rewrite the IFS recursion as:

$$w_k = h_{U_k}(w_{k-1}; \mathbf{S}_n), \tag{6}$$

where $U_k$ is a random variable taking values in $\{1, 2, \ldots, m_b\}$ and $p_i := \mathbb{P}(U_k = i)$. If the mini-batch sampling is uniform (i.e., the default option in practice), we have $p_i = 1/m_b$; however, we are not restricted to this option, the sampling scheme is allowed to be more general. We finally call the triple $(\mathbb{R}^d, \{h_i(\cdot; \mathbf{S}_n)\}_{i=1}^{m_b}, \{p_i\}_{i=1}^{m_b})$ an iterated function system (IFS).

Given a dataset $\mathbf{S}_n$, we are interested in the limiting behavior of the training algorithm (6). We characterize this behavior by considering the invariant measure $\mu_{W|\mathbf{S}_n}$ of the IFS (also called stationary distribution), that is a Borel probability measure on $\mathbb{R}^d$, such that $w_\infty \sim \mu_{W|\mathbf{S}_n}$. To be able to work in this context, we first need to ensure that the recursion (6) admits an invariant measure, i.e., $\mu_{W|\mathbf{S}_n}$ exists. Accordingly, we require the following mild conditions on the IFS (6). Let $U$ be a random variable with the same distribution as $U_k$. If the recursion (6) is *Lipschitz on average*, i.e.,

$$\mathbb{E}[L_U \mid \mathbf{S}_n] < \infty, \quad \text{with} \quad L_U := \sup_{x,y \in \mathbb{R}^d} \frac{\|h_U(x; \mathbf{S}_n) - h_U(y; \mathbf{S}_n)\|}{\|x - y\|}, \tag{7}$$

and is *contractive on average*, i.e., if

$$\mathbb{E}\left[\log(L_U) \mid \mathbf{S}_n\right] = \sum_{i=1}^{m_b} p_i \log(L_i) < 0, \qquad \text{with} \quad p_i > 0, \text{ for any } i = 1, \ldots, m_b, \tag{8}$$

then it can be shown that the process is ergodic and admits a unique invariant measure where the limit $\rho := \lim_{k \to \infty} (1/k) \log \|h_{U_k} h_{U_{k-1}} \cdots h_{U_1}\|$ exists almost surely and is a constant [Elt90], where $\rho$ is called the *Lyapunov exponent*. Furthermore, under further technical assumptions, it can be shown that (6) is geometrically ergodic [DF99]. We note that this condition for the existence of the invariant measure is only applicable to stochastic optimization algorithms with a constant stepsize in which case the random map $h_U$ is not time-varying. If decaying stepsize is used instead, then the limit may degenerate to be a singleton. For example, in the toy example illustrated in Figure 1 with quadratics in dimension one, if we use SGD with decaying stepsize $\eta_k = c/k$ where the positive constant $c$ is small enough, then the limit of the iterates will be a singleton as the iterates will converge to the global minimum of the optimization objective (see e.g. [GOP21, GOP19]).

Our goal will be to relate the generalization error to $\overline{\dim}_H \mu_{W|\mathbf{S}_n}$. To achieve this goal, at first sight, it might seem tempting to extract a full-measure set $A$ by using the definition of $\mu_{W|\mathbf{S}_n}$, such that $\mu_{W|\mathbf{S}_n}(A) = 1$ and $\dim_H A \approx \overline{\dim}_H \mu_{W|\mathbf{S}_n}$, and then directly invoke the results from [ŞSDE20], which would link the generalization error to $\dim_H A$, hence, also to $\overline{\dim}_H \mu_{W|\mathbf{S}_n}$. However, since [ŞSDE20] does not consider an IFS framework, the conditions they require (e.g., boundedness of $A$, $\dim_M A = \dim_H A$) are not suited to IFSs, and hence prevent us from directly using their results.

As a remedy, we make a detour and show that we can find *almost full-measure* sets $A$, such that $\mu_{W|\mathbf{S}_n}(A) \approx 1$ and $\overline{\dim}_M A \approx \overline{\dim}_H \mu_{W|\mathbf{S}_n}$ (notice that in this case we directly use the Minkowski dimension of $A$, as opposed to its Hausdorff dimension). To achieve this goal, we require the following geometric regularity condition on the invariant measure.

**H 1.** *For $\pi^\infty$-almost all $\mathbf{S}$ and all $n \in \mathbb{N}_+$, the recursion* (6) *satisfies* (7) *and* (8) *and the limit* $\lim_{r\to 0}\left[\log \mu_{W|\mathbf{S}_n}(B_d(w,r))\,/\log r\right]$ *exists for $\mu_{W|\mathbf{S}_n}$-almost every $w$.*

This is a common condition [Pes08] and is satisfied for a large class of measures. For instance, 'sufficiently regular' measures with the property that $C_1 r^s \leq \mu(B(x,r)) \leq C_2 r^s$ for some constant $s > 0$ and positive constants $C_1$, $C_2$ will satisfy this assumption. Such measures are called Ahlfors-regular (cf. [ŞSDE20, Assumption H4] for a related condition), and it is known that IFSs that satisfy certain 'open set conditions' lead to Ahlfors regular invariant measures (see [MT10, Section 8.3]). Yet, our assumption is more general and does not immediately require Ahlfors-regularity.

Under **H**1, we now formalize our key observation, which serves as the basis for our bounds.

**Proposition 1.** *Assume that **H**1 holds. Then for every $\varepsilon > 0$, $n \in \mathbb{N}_+$, and $\pi^\infty$-almost every $\mathbf{S}$, there exists $\delta := \delta(\varepsilon, \mathbf{S}_n) \in (0, 1]$ and a bounded measurable set $A_{\mathbf{S}_n,\delta} \subset \mathbb{R}^d$, such that*

$$\mu_{W|\mathbf{S}_n}(A_{\mathbf{S}_n,\delta}) \geq 1 - \delta, \quad and \quad \overline{\dim}_{\mathrm{M}} A_{\mathbf{S}_n,\delta} \leq \overline{\dim}_{\mathrm{H}}\mu_{W|\mathbf{S}_n} + \varepsilon, \tag{9}$$

*and $\delta(\varepsilon, \mathbf{S}_n) \to 0$ as $\varepsilon \to 0$.*

Thanks to this result, we can now leverage the proof technique presented in [ŞSDE20, Theorem 2], and link the generalization error to $\overline{\dim}_{\mathrm{H}}\mu_{W|\mathbf{S}_n}$ through $\overline{\dim}_{\mathrm{M}} A_{\mathbf{S}_n,\delta}$. We shall emphasize that, mainly due to the sets $A_{\mathbf{S}_n,\delta}$ not being of full-measure, our framework introduces additional non-trivial technical difficulties that we need to tackle in our proof.

We now introduce our second assumption, which roughly corresponds to a 'topological stability' condition, and is adapted from [ŞSDE20, Assumption H5]. Formally, consider the (countably infinite) collection of closed balls of radius $\beta$, whose centers are on the fixed grid $N_\beta := \left\{\left(\frac{(2j_1+1)\beta}{2\sqrt{d}}, \ldots, \frac{(2j_d+1)\beta}{2\sqrt{d}}\right) : j_i \in \mathbb{Z}, i = 1, \ldots, d\right\}$, and for a set $A \subset \mathbb{R}^d$, define $N_\beta(A) := \{x \in N_\beta : B_d(x, \beta) \cap A \neq \emptyset\}$, which is the collection of the centers of the balls that intersect $A$.

**H 2.** *Let $\mathcal{Z}^\infty := (\mathcal{Z} \times \mathcal{Z} \times \cdots)$ denote the countable product endowed with the product topology and let $\mathfrak{B}$ be the Borel $\sigma$-algebra generated by $\mathcal{Z}^\infty$. For a Borel set $A \subset \mathbb{R}^d$, let $\mathfrak{F}, \mathfrak{G}$ be the sub-$\sigma$-algebras of $\mathfrak{B}$ generated by the collections of random variables given by $\{\hat{\mathcal{R}}(w, \mathbf{S}_n) : w \in \mathbb{R}^d, n \geq 1\}$ and $\left\{\mathbb{1}\{w \in N_\beta(A_{\mathbf{S}_n,\delta})\}, \mu_{W|\mathbf{S}_n}(A_{\mathbf{S}_n,\delta}), \overline{\dim}_{\mathrm{H}}\mu_{W|\mathbf{S}_n} : \delta, \beta \in \mathbb{Q}_{>0}, w \in N_\beta, n \geq 1\right\}$, where $A_{\mathbf{S}_n,\delta}$ is given in Proposition 1. There exists a constant $M \geq 1$ such that for any $F \in \mathfrak{F}$, $G \in \mathfrak{G}$, we have $\mathbb{P}(F \cap G) \leq M\mathbb{P}(F)\mathbb{P}(G)$.*

**H**2 simply ensures that the dependence between the training error and the topological properties of the support of $\mu_{W|\mathbf{S}_n}$ can be controlled via $M$. Hence, it can be seen as a form of *algorithmic stability* [BE02], where $M$ measures the level of stability of the topology of $\mu_{W|\mathbf{S}_n}$: a small $M$ indicates that the geometrical structure of $\mu_{W|\mathbf{S}_n}$ does not heavily depend on the particular value of $\mathbf{S}_n$. The constant $M$ is also related to the mutual information [XR17, AAV18, HŞKM21], but may be better behaved than the mutual information as it relies on very specific functions of the random variables. Similar to mutual information, a-priori there is no reason to expect $M$ to be finite for general algorithms; intuitively, however, the more stochasticity an algorithm incorporates the more we expect the set $A_{\mathbf{S}_n,\delta}$ and the loss landscape to decouple. For example, for a purely random algorithm (which ignores $\mathbf{S}_n$) the two objects will be independent. In the other extreme, where the algorithm is deterministic given the sample may fail to be finite. Since we are controlling the generalization error on the support, which is itself a random set depending on the sample, we require **H**2 to be able to make progress.

We require one final assumption, which states that the loss $\ell$ is sub-exponential.

**H 3.** *$\ell$ is $L$-Lipschitz continuous in $w$, and when $z \sim \pi$, for all $w$, $\ell(w, z)$ is $(\nu, \kappa)$-sub-exponential, that is, for all $|\lambda| < 1/\kappa$, we have $\log \mathbb{E}_{z\sim\pi}\left[\exp\left(\lambda\left(\ell(w, z) - \mathcal{R}(w)\right)\right)\right] \leq \nu^2\lambda^2/\kappa$.*

Armed with these assumptions, we can now present our main result.

**Theorem 1.** *Assume that **H**1 to 3 hold and $\overline{\dim}_{\mathrm{H}}\mu_{W|\mathbf{S}_n} = \omega(\log\log(n)/\log(n))$, $\pi^\infty$-almost-surely. Then, the following bound holds for sufficiently large $n$:*

$$|\hat{\mathcal{R}}(W, \mathbf{S}_n) - \mathcal{R}(W)| \leq 8\nu\sqrt{\frac{\overline{\dim}_{\mathrm{H}}\mu_{W|\mathbf{S}_n}\log^2(nL^2)}{n}} + \frac{\log(13M/\zeta)}{n}, \tag{10}$$

*with probability at least $1 - 2\zeta$ over the joint distribution of $\mathbf{S} \sim \pi^\infty$, $W \sim \mu_{W|\mathbf{S}_n}$.*

This theorem shows that the Hausdorff dimension of the invariant measure acts as a 'capacity metric' and the generalization error is therefore directly linked to this metric, i.e., the complexity of the underlying fractal structure has close links to the generalization performance. On the other hand, the condition $\overline{\dim}_H \mu_{W|\mathbf{S}_n} = \omega(\log\log(n)/\log(n))$ is very mild and makes sure that the dimension of the IFS does not decrease very rapidly with increasing number of data points $n$. We shall mention that Theorem 1 has an asymptotic nature as we do not have an explicit control on how large $n$ should be. This is due to the fact that the notions of Minkowski and Hausdorff dimensions are essentially asymptotic, which unfortunately prevents us from obtaining any truly non-asymptotic result. However, obtaining nonasymptotic results might be possible with further assumptions on the fractal dimension of the invariant measures and their supports.

Theorem 1 enables us to access the rich theory of IFSs, where bounds on the Hausdorff dimension are readily available, and connect them to statistical learning theory. The following result is a direct corollary to Theorem 1 and [Ram06, Theorem 2.1] (see Theorem S2 in the supplementary document).

**Corollary 1.** *Assume that the conditions of Theorem 1 hold. Furthermore, consider the recursion (6) and assume that $h_i$ are continuously differentiable with derivatives $J_{h_i}$ that are $\alpha$-Hölder continuous for some $\alpha > 0$. Then, there exists a constant $M > 1$ such that for sufficiently large $n$:*

$$|\hat{\mathcal{R}}(W, \mathbf{S}_n) - \mathcal{R}(W)| \leq 8\nu \sqrt{\frac{\mathcal{E} \log^2(nL^2)}{\left[\sum_{i=1}^{m_b} p_i \int_{\mathbb{R}^d} \log(\|J_{h_i}(w)\|)\mathrm{d}\mu_{W|\mathbf{S}_n}(w)\right]n} + \frac{\log(13M/\zeta)}{n}}, \quad (11)$$

*with probability $1 - 2\zeta$ over $\mathbf{S} \sim \pi^\infty$, $W \sim \mu_{W|\mathbf{S}_n}$, where $\mathcal{E} := \sum_{i=1}^{m_b} p_i \log(p_i)$ denotes the negative entropy of the mini-batch sampling scheme, $\|\cdot\|$ denotes the operator norm (with the $\ell_2$-norm being the underlying norm), and $J_{h_i}$ is the Jacobian of $h_i$ defined in the notation section.*

Note that the Hölder condition is mainly used to ensure that the invariant measure exists and the constant $\alpha$ does not directly interact with the bound. However, it might affect the rate of convergence to the invariant measure.

By this result, we discover an interesting quantity, $\sum_{i=1}^{m_b} p_i \int_{\mathbb{R}^d} \log(\|J_{h_i}(w)\|)\mathrm{d}\mu_{W|\mathbf{S}_n}(w)$, which *simultaneously* captures the effects of the data and the algorithm[2]. To see it more clearly, let us consider the SGD recursion (3), where $\|J_{h_i}(w)\| = \|I - \eta\nabla^2\tilde{\mathcal{R}}_{S_i}(w)\|$ and $\{S_i\}_{i=1}^{m_b}$ denotes the enumeration of the mini-batches. Then, the overall quantity becomes

$$\mathbb{E}_{U,W}\left[\log\|I - \eta\nabla^2\tilde{\mathcal{R}}_{S_U}(W)\|\right], \quad (12)$$

where the expectation is taken over the mini-batch index $U \in \{1, \ldots, m_b\}$ with $\mathbb{P}(U = i) = p_i$, and $W \sim \mu_{W|\mathbf{S}_n}$. We clearly observe that this term depends on (i) the algorithm choice through the form of $h_i$, (ii) step-size $\eta$, (iii) batch-size through $m_b$, (iv) problem geometry through $\nabla^2\tilde{\mathcal{R}}$, and (v) data distribution through $\mu_{W|\mathbf{S}_n}$. We believe that such a compact representation of all these constituents and their interaction is novel and will open up interesting future directions.

## 4    Analytical Estimates for the Hausdorff Dimension

The generalization bound presented in Theorem 1 applies to a number of stochastic optimization algorithms that can be represented with an IFS and to a large class of losses that can be non-convex or convex. It is controlled by the Hausdorff dimension of the invariant measure $\mu_{W|\mathbf{S}_n}$ which needs to be estimated. In the numerical experiments section, we will discuss how this quantity can be estimated from the dataset $\mathbf{S}_n$ and the iterates of the underlying algorithm.

Corollary 1 shows that for smooth losses, the Hausdorff dimension can be controlled with the expectation of the norm of the logarithm of the Jacobian $\log(\|J_{h_i}(w)\|)$ with respect to the invariant measure $\mu_{W|\mathbf{S}_n}$. In general, an explicit characterization of the invariant measure is not known. Nevertheless, under additional appropriate assumptions that can hold in practice, such as boundedness of the data of the loss, we next discuss that it is possible to get uniform lower and upper bounds on the quantity $\|J_{h_i}(w)\|$ which leads to analytical upper bounds on $\overline{\dim}_H \mu_{W|\mathbf{S}_n}$.

---

[2]Note that thanks to [Ram06], we can allow state-dependent $p_i = p_i(w)$; yet, we do not consider this option as its application is not immediately clear.

As illustrative examples; in the following, we will consider the setting where we divide $\mathbf{S}_n$ into $m_b = m_b^{\text{batch}} = n/b$ batches with each batch having $b$ elements, and then we discuss how analytical estimates on the (upper) Hausdorff dimension $\overline{\dim}_{\mathrm{H}}\mu_{W|\mathbf{S}_n}$ can be obtained for some particular problems including least squares, regularized logistic regression, and one hidden-layer networks. In the supplementary document, we also discuss how similar bounds can be obtained for support vector machines and other algorithms such as preconditioned SGD and stochastic Newton methods.

**Least squares.** We consider the least squares problem, with data points $z_i = (a_i, y_i)$ and loss

$$\ell(w, z_i) := \left(a_i^T w - y_i\right)^2 / 2 + \lambda \|w\|^2 / 2, \tag{13}$$

where $\lambda > 0$ is a regularization parameter.

**Proposition 2** (Least squares)**.** *Consider the least squares problem* (13)*. Assume the step-size $\eta \in (0, \frac{1}{R^2 + \lambda})$, where $R := \max_i \|a_i\|$ is finite. Then, we have the following upper bound:*

$$\overline{\dim}_{\mathrm{H}}\mu_{W|\mathbf{S}_n} \leq \frac{\log (n/b)}{\log(1/(1 - \eta\lambda))}. \tag{14}$$

Note that here $\ell$ is only pseudo-Lipschitz $|\ell(w, z_i) - \ell(w', z_i)| \leq L_i(1 + \|w\| + \|w'\|)\|w - w'\|$ for some $L_i > 0$, rather than globally Lipschitz. However; the conditions in Proposition 2 ensure that $w$ will stay in a bounded region, in which case $\ell$ becomes Lipschitz. Also note that only the logarithm of the Lipschitz constant directly enters the bound.

We observe that, for fixed $n$, the upper bound for $\overline{\dim}_{\mathrm{H}}\mu_{W|\mathbf{S}_n}$ is decreasing both in $\eta$ and $b$. This behavior is not surprising: large $\eta$ results in chaotic behaviors (cf. Figures 1,2), and in the extreme case where $b = n$, the algorithm becomes deterministic and hence converges to a single point, in which case the Hausdorff dimension becomes 0. However, the decrement due to $b$ does not automatically grant good generalization performance: since the algorithm becomes deterministic, the stability constant $M$ in **H**2 can get arbitrarily large, hence the bound in Theorem 1 could become vacuous. This outcome reveals an interesting tradeoff between the Hausdorff dimension and the constant $M$, through the batch-size $b$, and investigating this tradeoff is an interesting future direction.

We further notice that the numerator in (14) is logarithmically increasing with $n$, which is compensated by the factor $1/n$ in Theorem 1. Nevertheless, we can take the batch-size in proportion with $n$ (i.e., setting $m_b$ to a constant value), in order avoid this logarithmic growth. We also note that the input dimension $d$ potentially affects the term $R$, which forms the bound for the input data, and hence the input dimension will indirectly affect the generalization bound. Finally, regarding the remaining bounds in this section, even though their forms might differ from (14), similar remarks also apply. Hence, we will omit the discussion.

**Regularized logistic regression.** Given the data points $z_i = (a_i, y_i)$, consider regularized logistic regression with the loss:

$$\ell(w, z_i) := \log \left(1 + \exp\left(-y_i a_i^T w\right)\right) + \lambda \|w\|^2 / 2, \tag{15}$$

where $\lambda > 0$ is a regularization parameter. We have the following result.

**Proposition 3** (Regularized logistic regression)**.** *Consider the regularized logistic regression* (15)*. Assume the step-size $\eta < 1/\lambda$ and the input data is bounded, i.e. $R := \max_i \|a_i\| < 2\sqrt{\lambda}$. We have:*

$$\overline{\dim}_{\mathrm{H}}\mu_{W|\mathbf{S}_n} \leq \frac{\log (n/b)}{\log(1/(1 - \eta\lambda + \frac{1}{4}\eta R^2))}. \tag{16}$$

Next, we consider a non-convex formulation for robust regression (see e.g. [MBM18]), with $z_i = (a_i, y_i)$ and the loss

$$\ell(w, z_i) := \rho\left(y_i - \langle w, a_i \rangle\right) + \lambda \|w\|^2 / 2, \tag{17}$$

where $\lambda > 0$ is a regularization parameter and $\rho$ is a non-convex function, assumed to be twice continuously differentiable, where a standard choice is *Tukey's bisquare loss* defined as $\rho_{\text{Tukey}}(t) = 1 - (1 - (t/t_0)^2)^3$ for $|t| \leq t_0$, and $\rho_{\text{Tukey}}(t) = 1$ for $|t| \geq t_0$ (see e.g. [MBM18]), and exponential squared loss: $\rho_{\exp}(t) = 1 - e^{-|t|^2/t_0}$, where $t_0 > 0$ is a tuning parameter (see e.g. [WJHZ13]).

**Proposition 4** (Non-convex formulation for robust regression). *Consider the non-convex formulation for robust regression* (17). *Assume the step-size* $\eta < \frac{1}{\lambda + R^2(2/t_0)}$, *where* $R = \max_i \|a_i\| < \sqrt{\lambda t_0/2}$. *Then, we have the following upper bound for the Hausdorff dimension:*

$$\overline{\dim}_{\mathrm{H}} \mu_{W|\mathbf{S}_n} \leq \frac{\log (n/b)}{\log(1/(1 - \eta\lambda + \eta R^2 \frac{2}{t_0}))}. \tag{18}$$

**One hidden-layer neural network.** Given the data points $z_i = (a_i, y_i)$. Let $a_i \in \mathbb{R}^d$ be the input and $y_i \in \mathbb{R}^m$ be the corresponding output, and let $w_r \in \mathbb{R}^d$ be the weights of the $r$-th hidden neuron of a one hidden-layer network, and $b_r \in \mathbb{R}$ is the output weight of hidden unit $r$. For simplicity of the presentation, following [ZMG19, DZPS19], we only optimize the weights of the hidden layer, i.e. $w = \begin{bmatrix} w_1^T w_2^T \dots w_m^T \end{bmatrix}$ is the decision variable with the regularized squared loss:

$$\ell(w, z_i) := \|y_i - \hat{y}_i\|^2 + \lambda\|w\|^2/2, \quad \hat{y}_i := \sum_{r=1}^m b_r \sigma\left(w_r^T a_i\right), \tag{19}$$

where the non-linearity $\sigma : \mathbb{R} \to \mathbb{R}$ is smooth and $\lambda > 0$ is a regularization parameter.

**Proposition 5** (One hidden-layer network). *Consider the one hidden-layer network* (19). *Assume the step-size* $\eta < \frac{1}{2\lambda}$. *Then, we have the following upper bound for the Hausdorff dimension:*

$$\overline{\dim}_{\mathrm{H}} \mu_{W|\mathbf{S}_n} \leq \frac{\log (n/b)}{\log(1/(1 - \eta(\lambda - C)))}, \tag{20}$$

*where* $C := M_y\|b\|_\infty\|\sigma''\|_\infty R^2 + \left(\max_j \|v_j\|_\infty\right)^2 < \lambda$, *where* $R := \max_i \|a_i\|$, $M_y := \max_i \|y_i - \hat{y}_i\|$, *and* $v_i := \begin{bmatrix} b_1\sigma'(w_1^T a_i)a_i \; b_2\sigma'(w_2^T a_i)a_i \; \cdots b_m\sigma'(w_m^T a_i)a_i \end{bmatrix}^T$.

In Proposition 4, we assumed that $C$ is uniformly bounded and $\lambda > C$ for mathematical convenience. However, in general, $C$ by its definition might increase in $m$ and $n$ so that we do not expect that we can choose small $\lambda$ for large data sets and wide networks.

## 5 Experiments

Our aim now is to empirically investigate whether the bound in Corollary 1 is informative, in that it is predictive of a neural network's generalization error. As the second term of this bound cannot be evaluated, we will assume that it is negligible and focus our efforts on the first term. Further, because the denominator of the first term is the only term that depends on the invariant measure, we want to establish that the inverse of this denominator is predictive of a neural network's generalization error. Note however that for complex models, such as modern neural networks, analytically bounding $\|J_{h_i(x)}\|$ becomes highly non-trivial.

In our experiments, we fix the algorithm to SGD and we develop a numerical method for computing the term (12). Noting that $J_{h_i}(w) = I - \eta\nabla^2\tilde{\mathcal{R}}_i(w)$, for simplicity we denote the inverse of (12) as the 'complexity': $R = 1/\left[\sum_{i=1}^{m_b} p_i \int_{\mathbb{R}^d} \log\left(\|J_{h_i}(w)\|\right) \mathrm{d}\mu_{W|\mathbf{S}_n}(w)\right]$[3].

To approximate the expectations, we propose the following simple Monte Carlo strategy:

$$R^{-1} \approx \left[1/(N_W N_U)\right] \sum_{i=1}^{N_W} \sum_{j=1}^{N_U} \log\left(\|J_{h_{U_j}}(W_i)\|\right), \tag{21}$$

where $U_j$ denotes i.i.d. random mini-batch indices that are drawn without-replacement from $\{1, \dots, n\}$ and $W_i \overset{\text{i.i.d.}}{\sim} \mu_{W|S}$. Assuming (8) is ergodic [DF99], we treat the iterates $w_k$ as i.i.d. samples from $\mu_{W|\mathbf{S}_n}$ for large $k$, hence, $\log(\|J_{h_{U_j}}(W_i)\|)$ can be computed on these iterates, and (21) can be computed accordingly. Our implementation for computing $\|J_{h_{U_j}}(W_i)\|$ for neural networks with millions of parameters is detailed in the supplementary document. Though the size of $J_{h_i}$ is very large in our experiments ($\approx 20\mathrm{M} \times 20\mathrm{M}$ on average), our algorithm can efficiently compute the norms without constructing $J_{h_{U_j}}(W_i)$, by extending the approach presented in [YGKM20]. Our implementation is available at `https://github.com/umutsimsekli/fractal_generalization`.

---

[3]Note that the first term of the bound in Corollary 1 suggests computing $\sqrt{R}$ rather than $R$; however, both choices yield very similar results with high statistical significance.

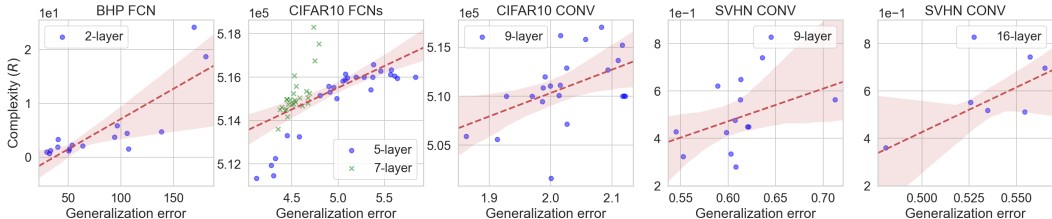

Figure 3: Estimates of $R$ plotted against the generalization error ($|\text{training loss} - \text{test loss}|$) for VGG11 and FCNs trained on CIFAR10, SVHN and BHP with varying $\eta, b$. The linear regression of best fit is plotted in red, where shading corresponds to the $95\%$ confidence interval. For all plots the one-sided p-value, testing whether the null hypothesis that the slope of the line is in-fact negative and not positive, is significantly less than $0.001$, indicating that it is highly likely that $R$ and the generalization error are positively correlated.

In Figure 3 we plot the estimates of $R$ for a variety of convolutional (CONV) and fully connected network (FCN) architectures trained on CIFAR10, SVHN and Boston House Prices (BHP). For the full details of the models, the hardware used, their run-time, and the convergence criterion used, see Section S7 in the supplement. The plot demonstrates that $R$ and generalization error are positively correlated and that this correlation is significant (p-value $\ll 0.001$) for all model architectures. This provides evidence that the bound on the generalization error in Corollary 1 is informative.

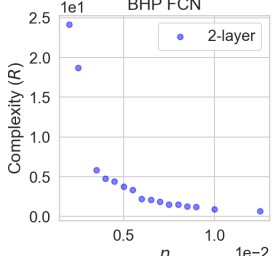

Figure 4: Estimated $R$ plotted against $\eta$ for 2 layer FCN trained on BHP.

To support our findings in Section 4 that the bound for the Hausdorff dimension $\overline{\dim}_H \mu_{W|\mathbf{S}_n}$ is monotonically decreasing in the step-size $\eta$, we plot $R$ against $\eta$ in Figure 4 for the networks trained on BHP in Figure 3. $R$ decreases with increasing $\eta$, clearly backing our findings.

We note that these results were inconclusive for classification models trained with a cross-entropy loss, in that we could not clearly observe a negative or positive correlation. Future work will further study this lack of correlation, particular to classification models.

## 6  Conclusion

In this work, we investigated stochastic optimization algorithms through the lens of IFSs and studied their generalization properties. Under some assumptions, we showed that the generalization error can be controlled based on the Hausdorff dimension of the invariant measure determined by the iterations, which can lead to tighter bounds than the standard bounds based on the ambient dimension. We proposed an efficient methodology to estimate the Hausdorff dimension in deep learning settings and supported our theory with several experiments on neural networks. We also illustrated our bounds on specific problems and algorithms such as SGD and its preconditioned variants, which unveil new links between generalization, algorithm parameters and the Hessian of the loss.

Our study does not have a direct societal impact as it is largely theoretical. The limitation of our study is its asymptotic nature due to operating on invariant measures. Future work will address (i) obtaining nonasymptotic bounds in terms of the number of iterations $k$, (ii) including the term of (12) as a regularizer to the optimization problem, which would be an alternative to the methods that aim to "decrease the intrinsic dimension" [ZQH$^+$18, BLGŞ21].

**Acknowledgements.** MAE is partially funded by CIFAR AI Chairs program, and CIFAR AI Catalyst grant, NSERC Grant [2019-06167]. MG's research is supported in part by the grants Office of Naval Research Award Number N00014-21-1-2244, National Science Foundation (NSF) CCF-1814888, NSF DMS-2053485, NSF DMS-1723085. UŞ's research is supported by the French government under management of Agence Nationale de la Recherche as part of the "Investissements d'avenir" program, reference ANR-19-P3IA-0001 (PRAIRIE 3IA Institute). LZ is grateful to the partial support from NSF DMS-2053454 and a Simons Foundation Collaboration Grant.

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
