# Fractal Structure and Generalization Properties of Stochastic Optimization Algorithms

## SUPPLEMENTARY DOCUMENT

This document provides additional material for the NeurIPS 2021 submission entitled *"Fractal Structure and Generalization Properties of Stochastic Optimization Algorithms"*. The document is organized as follows:

- Technical background for the proofs.
  - In Section S1, we provide additional background on dimension theory. In particular we define the Minkowski dimension and the local dimension for a *measure*. Then, we provide three existing theoretical results that will be used in our proofs.
- Additional theoretical results.
  - In Section S2, we provide an upper-bound on the Hausdorff dimension of the invariant measure of SGD, when applied on support vector machines. This result is a continuation of the results given in Section 4.
  - In Section S3, we provide upper-bounds on the Hausdorff dimension of the invariant measure of *preconditioned* SGD on different problems.
  - In Section S4, we provide an upper-bound on the Hausdorff dimension of the invariant measure of the *stochastic Newton algorithm* applied on linear regression.
  - In Section S5, we illustrate the conditions (7) and (8) on a simple setup.
- Details of the experimental results.
  - In Section S6, we provide the details of the algorithm that we developed for computing the operator norm $\|I - \eta \nabla^2 \tilde{R}_k(w)\|$ for neural networks.
  - In Section S7, we provide the details of the SGD hyperparameters, network architectures, and information regarding hardware/run-time.
- Proofs.
  - In Section S8, we provide the proofs all the theoretical results presented in the main document and the supplementary document.

## S1 Further Background on Dimension Theory

### S1.1 Minkowski dimension of a measure

Based on the definition of the Minkowski dimension of sets as given in Section 2, we can define the Minkowski dimension of a finite Borel measure $\mu$, as follows [Pes08]:

$$\overline{\dim}_{\mathrm{M}}\mu := \lim_{\delta \to 0} \inf \left\{ \overline{\dim}_{\mathrm{M}}Z : \mu(Z) \geq 1 - \delta \right\}. \tag{S1}$$

Note that in general, we have

$$\overline{\dim}_{\mathrm{M}}\mu \leq \inf \left\{ \overline{\dim}_{\mathrm{M}}Z : \mu(Z) = 1 \right\},$$

where the inequality can be strict, see [Pes08, Chapter 7].

### S1.2 Local dimensions of a measure

It is sometimes more convenient to consider a dimension notion that is defined in a pointwise manner. Let $\mu$ be a finite Borel regular measure on $\mathbb{R}^d$. The lower and upper local (or pointwise) dimensions of $\mu$ at $x \in \mathbb{R}^d$ are respectively defined as follows:

$$\underline{\dim}_{\mathrm{loc}}\mu(x) := \liminf_{r \to 0} \frac{\log \mu(B(x,r))}{\log r}, \tag{S2}$$

$$\overline{\dim}_{\mathrm{loc}}\mu(x) := \limsup_{r \to 0} \frac{\log \mu(B(x,r))}{\log r}, \tag{S3}$$

where $B(x, r)$ denotes the ball with radius $r$ about $x$. When the values of these dimensions agree, the common value is called the local (or pointwise) dimension of $\mu$ at $x$, and is denoted by $\dim_{\text{loc}} \mu(x)$. The local dimensions describe the power-law behavior of $\mu(B(x, r))$ for small $r$ [Fal97]. These dimensions are closely linked to the Hausdorff dimension.

## S1.3 Existing Results

The following result from [Ram06] upper-bounds the Hausdorff dimension of the invariant measure of an IFS to the constituents of the IFS. We translate the result to our notation. Before we proceed, let us first introduce open set conditions from [Ram06]. The IFS satisfies the *open set condition* (OSC) if the IFS is contracting, and there exists an open set $U$ such that $h_i(U) \subset U$ and $h_i(U) \cap h_j(U) = \emptyset$ for $i \neq j$. The IFS satisfies the *strong open set condition* (SOSC) if the IFS satisfies OSC for some open set $U$ and if there exists some $R_1 > 0$ such that $\text{dist}(h_i(U), h_j(U)) \geq R_1$. The IFS satisfies the *regular open set condition* (ROSC) if the IFS satisfies OSC for some open set $U$ and in addition there exist some $R_2, R_3 > 0$ such that $\text{vol}(B_r(x) \cap U) \geq R_3 r^d$ for any $r < R_2$, $x \in U$. We have the following result.

**Theorem S2.** *[Ram06, Theorem 2.1] Consider the IFS (6) and assume that conditions (7) and (8) are satisfied and $h_i$ are continuously differentiable with derivatives $J_{h_i}$ that are $\alpha$-Hölder continuous for some $\alpha > 0$. The invariant measure $\mu_{W|\mathbf{S}_n}$ of the IFS satisfies*

$$\overline{\dim}_{\text{H}} \mu_{W|\mathbf{S}_n} \leq s \quad where \quad s := \frac{\mathcal{E}}{\sum_{i=1}^{m_b} p_i \int_{x \in \mathbb{R}^d} \log(\|J_{h_i}(w)\|) d\mu_{W|\mathbf{S}_n}(w)},$$

*where $\mathcal{E} := \sum_{i=1}^{m_b} p_i \log(p_i)$ is the (negative) entropy. Furthermore, if $h_i$ are conformal and either SOSC or ROSC is satisfied, then we have*

$$\dim_{\text{H}}(\mu_{W|\mathbf{S}_n}) = \underline{\dim}_{\text{H}}(\mu_{W|\mathbf{S}_n}) = s.$$

The next two results link the Hausdorff and Minkowski dimensions of a measure to its local dimension.

**Proposition S6.** *[Fal97, Propositions 10.3] For a finite Borel measure $\mu$, the following identity holds:*

$$\overline{\dim}_{\text{H}} \mu = \inf \left\{ s : \underline{\dim}_{\text{loc}} \mu(x) \leq s \text{ for } \mu\text{-almost all } x \right\}. \tag{S4}$$

**Theorem S3.** *[Pes08, Theorem 7.1] Let $\mu$ be a finite Borel measure on $\mathbb{R}^d$. If $\overline{\dim}_{\text{loc}} \mu(x) \leq \alpha$ for $\mu$-almost every $x$, then $\overline{\dim}_{\text{M}} \mu \leq \alpha$.*

The next theorem, called Egoroff's theorem, will be used in our proofs repeatedly. It provides a condition for measurable functions to be uniformly continuous in an almost full-measure set.

**Theorem S4** (Egoroff's Theorem). *[Bog07, Theorem 2.2.1] Let $(X, \mathcal{A}, \mu)$ be a space with a finite nonnegative measure $\mu$ and let $\mu$-measurable functions $f_n$ be such that $\mu$-almost everywhere there is a finite limit $f(x) := \lim_{n \to \infty} f_n(x)$. Then, for every $\varepsilon > 0$, there exists a set $X_\varepsilon \in \mathcal{A}$ such that $\mu(X \backslash X_\varepsilon) < \varepsilon$ and the functions $f_n$ converge to $f$ uniformly on $X_\varepsilon$.*

## S2 Additional Analytical Estimates for SGD

**Support vector machines.** Given the data points $z_i = (a_i, y_i)$ with the input data $a_i$ and the output $y_i \in \{-1, 1\}$, consider support vector machines with smooth hinge loss:

$$\ell(w, z_i) := \ell_\sigma \left( y_i a_i^T w \right) + \lambda \|w\|^2 / 2, \tag{S5}$$

where $\sigma > 0$ is a smoothing parameter, $\lambda > 0$ is the regularization parameter and $\ell_\sigma(z) := 1 - z + \sigma \log(1 + e^{-\frac{1-z}{\sigma}})$. This loss function is a smooth version of the hinge loss that can be easier to optimize in some settings. In fact, it can be shown that as $\sigma \to 0$, this loss converges to the (non-smooth) hinge loss pointwise.

**Proposition S7** (Support vector machines). *Consider the support vector machines (S5). Assume the step-size $\eta < \frac{1}{\lambda + \|R\|^2/(4\rho)}$, where $R := \max_i \|a_i\|$ is finite. Then, we have:*

$$\overline{\dim}_{\text{H}} \mu_{W|\mathbf{S}_n} \leq \frac{\log(n/b)}{\log(1/(1 - \eta\lambda))}. \tag{S6}$$

## S3 Analytical Estimates for Preconditioned SGD

We consider the pre-conditioned SGD methods

$$w_k = w_{k-1} - \eta H^{-1} \nabla \tilde{R}_k(w_{k-1}), \tag{S7}$$

for a fixed square matrix $H$. Some choices of $H$ includes a diagonal matrix, a block diagonal matrix or the Fisher-information matrix (see e.g. [ZMG19]). We assume that $H$ is a positive definite matrix, and by Cholesky decomposition, we can write $H = SS^T$, where $S$ is a real lower triangular matrix with positive diagonal entries. If we have $H = JJ^T$, where $J$ is the Jacobian, then the corresponding least square problems is called the Gauss-Newton methods for least squares. Assume that $H$ there exist some $m, M > 0$ such that:

$$0 \prec mI \preceq H \preceq MI. \tag{S8}$$

As illustrative examples; in the following, we will consider the setting where we divide $\mathbf{S}_n$ into $m_b = n/b$ batches with each batch having $b$ elements, and then we discuss how analytical estimates on the (upper) Hausdorff dimension $\overline{\dim}_H \mu_{W|\mathbf{S}_n}$ can be obtained for some particular problems including least squares, regularized logistic regression, support vector machines, and one hidden-layer network.

**Least squares.** We consider the least square problem with data points $z_i = (a_i, y_i)$ and the loss

$$\ell(w, z_i) := \frac{1}{2}\left(a_i^T w - y_i\right)^2 + \frac{\lambda}{2}\|w\|^2, \tag{S9}$$

where $\lambda > 0$ is a regularization parameter. If we apply preconditioned SGD this results in the recursion (6) with

$$h_i(w) = M_i w + q_i \quad \text{with} \quad M_i := I - \eta\lambda H^{-1} - \eta H^{-1} H_i, \tag{S10}$$

$$H_i := \frac{1}{b}\sum_{j \in S_i} a_j a_j^T, \quad q_i := \frac{\eta}{b}H^{-1}\sum_{j \in S_i} a_j y_j,$$

where $a_j \in \mathbb{R}^d$ are the input vectors, and $y_j$ are the output variables, and $\{S_i\}_{i=1}^{m_b}$ is a partition of $\{1, 2, \ldots, n\}$ with $|S_i| = b$, where $i = 1, 2, \ldots, m_b$ with $m_b = n/b$. We have the following result.

**Proposition S8** (Least squares). *Consider the pre-conditioned SGD method (S7) for the least square problem (S9). Assume that the step-size $\eta < \frac{m}{R^2+\lambda}$, where $R := \max_i \|a_i\|$ is finite. Then, we have the following upper bound for the Hausdorff dimension:*

$$\overline{\dim}_H \mu_{W|\mathbf{S}_n} \leq \frac{\log(n/b)}{\log(1/(1 - \eta M^{-1}\lambda))}. \tag{S11}$$

**Regularized logistic regression.** We consider the regularized logistic regression problem with the data points $z_i = (a_i, y_i)$ and the loss:

$$\ell(w, z_i) := \log\left(1 + \exp\left(-y_i a_i^T w\right)\right) + \frac{\lambda}{2}\|w\|^2, \tag{S12}$$

where $\lambda > 0$ is the regularization parameter.

**Proposition S9** (Regularized logistic regression). *Consider the pre-conditioned SGD method (S7) for regularized logistic regression (S12). Assume that the step-size $\eta < m/\lambda$ and $R := \max_i \|a_i\| < 2\sqrt{m\lambda/M}$. Then, we have the following upper bound for the Hausdorff dimension:*

$$\overline{\dim}_H \mu_{W|\mathbf{S}_n} \leq \frac{\log(n/b)}{\log(1/(1 - \eta M^{-1}\lambda + \frac{1}{4}\eta m^{-1}R^2))}. \tag{S13}$$

Next, we consider a non-convex formulation for robust regression. Consider the data points $z_i = (a_i, y_i)$ and the loss:

$$\ell(w, z_i) := \rho\left(y_i - \langle w, a_i \rangle\right) + \frac{\lambda}{2}\|w\|^2, \tag{S14}$$

where $\lambda > 0$ is a regularization parameter and $\rho$ is a non-convex function. We have the following result.

**Proposition S10** (Non-convex formulation for robust regression). *Consider the pre-conditioned SGD method* (S7) *in the non-convex robust regression setting* (S14). *Assume that the step-size* $\eta < \frac{m}{\lambda + R^2(2/t_0)}$ *and* $R := \max_i \|a_i\| < \sqrt{\lambda t_0 m/(2M)}$. *Then, we have the following upper bound for the Hausdorff dimension:*

$$\overline{\dim}_{\mathrm{H}} \mu_{W|\mathbf{S}_n} \le \frac{\log(n/b)}{\log(1/(1 - \eta M^{-1}\lambda + \eta m^{-1} R^2 \frac{2}{t_0}))}. \tag{S15}$$

**Support vector machines.** We have the following result for pre-conditioned SGD when applied to the support vector machines problem (S5).

**Proposition S11** (Support vector machines). *Consider pre-conditioned SGD* (S7) *for support vector machines* (S5). *Assume that the step-size* $\eta < \frac{m}{\lambda + \|R\|^2/(4\rho)}$ *where* $R := \max_i \|a_i\|$ *is finite. Then, we have the following upper bound for the Hausdorff dimension:*

$$\overline{\dim}_{\mathrm{H}} \mu_{W|\mathbf{S}_n} \le \frac{\log(n/b)}{\log(1/(1 - \eta M^{-1}\lambda))}. \tag{S16}$$

**One hidden-layered neural network.** Consider the one-hidden-layer neural network setting as in Proposition 5, where the objective is to minimize the regularized squared loss with the loss function:

$$\ell(w, z_i) := \|y_i - \hat{y}_i\|^2 + \frac{\lambda}{2}\|w\|^2, \quad \hat{y}_i := \sum_{r=1}^m b_r \sigma\left(w_r^T a_i\right), \tag{S17}$$

where the non-linearity $\sigma : \mathbb{R} \to \mathbb{R}$ is smooth and $\lambda > 0$ is the regularization parameter.

**Proposition S12** (One hidden-layer network). *Consider the one-hidden-layer network* (S17). *Assume that* $\eta < \frac{m}{C+\lambda}$ *and* $\lambda > \frac{M}{m}C$, *where* $C$ *is defined in Corollary 5. Then, we have the following upper bound for the Hausdorff dimension:*

$$\overline{\dim}_{\mathrm{H}} \mu_{W|\mathbf{S}_n} \le \frac{\log(n/b)}{\log(1/(1 - \eta(M^{-1}\lambda - m^{-1}C)))}. \tag{S18}$$

## S4 Analytical Estimates for Stochastic Newton

We consider the stochastic Newton method

$$w_k = w_{k-1} - \eta \left[\tilde{H}_k(w_{k-1})\right]^{-1} \nabla \tilde{\mathcal{R}}_k(w_{k-1}), \quad \text{where} \quad \tilde{H}_k(w) := (1/b)\sum_{i \in \Omega_k} \nabla^2 \ell(w, z_i),$$

see e.g. [RKM16], where $\Omega_k = S_i$ with probability $p_i$ with $i = 1, 2, \ldots, m_b$, where $m_b = n/b$.

For simplicity, we focus on the least square problem, with the data points $z_i = (a_i, y_i)$ and the loss:

$$\ell(w, z_i) := \frac{1}{2}\left(a_i^T w - y_i\right)^2 + \frac{\lambda}{2}\|w\|^2, \tag{S19}$$

where $\lambda > 0$ is a regularization parameter. If we apply stochastic Newton this results in the recursion (6) with

$$h_i(w) = M_i w + q_i \quad \text{with} \quad M_i := (1 - \eta)I, \tag{S20}$$

$$\tilde{H}_i := \frac{1}{b}\sum_{j \in S_i} a_j a_j^T + \lambda I, \quad q_i := \frac{\eta}{b}\tilde{H}_i^{-1}\sum_{j \in S_i} a_j y_j,$$

where $a_j \in \mathbb{R}^d$ are the input vector, and $y_j$ are the output variable, and $\{S_i\}_{i=1}^{m_b}$ is a partition of $\{1, 2, \ldots, n\}$ with $|S_i| = b$, where $i = 1, 2, \ldots, m_b$ with $m_b = n/b$. Therefore, $J_{h_i}(w) = (1 - \eta)I$. By following the similar argument as in the proof of Proposition S8, we conclude that for any $\eta \in (0, 1)$,

$$\overline{\dim}_{\mathrm{H}} \mu_{W|\mathbf{S}_n} \le \frac{\log(n/b)}{\log(1/(1 - \eta))}, \tag{S21}$$

where the upper bound is decreasing in step-size $\eta$ and batch-size $b$.

## S5  Illustration of the Conditions (7) and (8)

In this section, we will demonstrate how to validate the conditions (7) and (8) on the least squares problem:

$$\hat{\mathcal{R}}(w) := \hat{\mathcal{R}}(w, \mathbf{S}_n) = \frac{1}{n} \sum_{i=1}^{n} \left(y_i - x_i^T w\right)^2,$$

where $x_i \in \mathbb{R}^d$'s are isotropic sub-Gaussian random vectors.

Consider a uniform subset $S \subset \{1, ..., n\}$ of size $b$. By the properties of sub-Gaussian random vectors (e.g. see [Ver18, Theorem 4.7.1] ), we have $\mathbb{E}[\|I - \frac{1}{b} \sum_{i \in S} x_i x_i^T\|] \leq CK^2 \sqrt{\frac{d}{b}}$ where $C$ is a positive constant, $K$ is the sub-Gaussian norm of $x_i$, which is true for a sufficiently large batch size. Now, consider the SGD update with mini-batch size $b$:

$$w_{k+1} = w_k - \frac{\eta}{b} \sum_{i \in S} x_i (x_i^T w - y_i) =: h_S(w_k, S).$$

We look at the Lipschitz constant of this function system: For any $S$,

$$\|h_S(w, S) - h_S(w', S)\| = \left\| w - w' - \frac{\eta}{b} \sum_{i \in S} x_i x_i^T (w - w') \right\| \leq \left\| I - \frac{\eta}{b} \sum_{i \in S} x_i x_i^T \right\| \|w - w'\|.$$

Notice that the quantity $\|I - \frac{\eta}{b} \sum_{i \in S} x_i x_i^T\|$ is an upper bound on the Lipschitz constant $L_S$ of $h_S$. Investigating the condition (8), we have $\mathbb{E}[\log(L_S)] \leq \log\left( \eta C K^2 \sqrt{\frac{d}{b}} + (1 - \eta) \right)$, where the first step follows from Jensen's inequality and the second follows from sub-Gaussian property. The right hand size of the above bound can be made smaller than 0 for a sufficiently small step size choice $\eta$.

## S6  Estimating the Complexity $R$ for SGD

Estimating $R$, as detailed in Equation (21), requires drawing $N_W$ samples from the invariant measure and $N_U$ batches of from the training data.

As mentioned in the main text, to approximate the summation over $N_W$ samples from the invariant measure, assuming (8) is ergodic [DF99], we treat the iterates $w_k$ as i.i.d. samples from $\mu_{W|S}$ for large $k$, hence, the norm of the Jacobian $\log(\|J_{h_{I_j}}(W_i)\|)$ can be efficiently computed on these iterates. Thus, we first we train a neural-network to convergence, whereby convergence is defined as the model reaching some accuracy level (if the dataset is a classification task) *and* achieving a loss below some threshold on training data. We assume that after convergence the SGD iterates will be drawn from the invariant measure. As such we run the training algorithm for another 200 iterates, saving a snapshot of the model parameters at each step, such that $N_W = 200$ in Equation (21). For each of these snapshots we estimate the spectral norm $\|J_{h_I}(W)\|$ using a simple modification of the power iteration algorithm of [YGKM20], detailed in Section S6.1 below. This modified algorithm is scalable to neural networks with millions of parameters and we apply it to 50 of the batches used during training, such that $N_U = 50$ in (21).

### S6.1  Power Iteration Algorithm for $\|J_{h_i}(w)\|$

We re-purpose the power iteration algorithm of [YGKM20] adding a small modification that allows for the estimation of the spectral norm $\|J_{h_i}(w)\|$. We first note that

$$J_{h_i}(w) = I - \eta \nabla^2 \tilde{\mathcal{R}}_i(w) \tag{S22}$$

where $\nabla^2 \tilde{\mathcal{R}}_i(w)$ is the Hessian for the $i^{th}$ batch. As such our power iteration algorithm needs to estimate the operator norm of the matrix $I - \eta \nabla^2 \tilde{\mathcal{R}}_i(w)$ and not just that of the Hessian of the network. To do this we just need to change the 'vector-product' step of the power-iteration algorithm of [YGKM20]. Our modified method has the same convergence guarantees, namely that the method

will converge to the 'true' top eigenvalue if this eigenvalue is 'dominant', in that it dominates all other eigenvalues in absolute value, i.e if $\lambda_1$ is the top eigenvalue then we must have that:

$$|\lambda_1| > |\lambda_2| \geq \ldots |\lambda_n|$$

to guarantee convergence.

---

**Algorithm 1:** Power Iteration for Top Eigenvalue Computation of $J_{h_i}(w)$

---

**Input:** Network Parameters: $w$, Loss function: $f$, Learning rate: $\eta$

1   Compute the gradient of $f$ by backpropagation, i.e., compute $g_w = \frac{df}{dw}$.
2   Draw a random vector $v$ from $N(0,1)$ (same dimension as $w$).
3   Normalize $v$, $v = \frac{v}{\|v\|_2}$
4   **for** $i = 1, 2, \ldots$ **do**           `// Power Iteration`
5      Compute $gv = g_w^T v$                      `// Inner product`
6      Compute $Hv$ by backpropagation, $Hv = \frac{d(gv)}{dw}$    `// Get Hessian vector product`
7      Compute $J_{h_i}v$, $J_{h_i}v = (I - \eta H)v = v - \eta Hv$     `// Get `$J_{h_i}$` vector product`
8      Normalize and reset $v$, $v = \frac{J_{h_i}v}{\|J_{h_i}v\|_2}$
9   **end**

---

## S7    Experiment Hyperparameters

**Training Parameters:**    All models in Figures 3 and 4 were trained using SGD with batch sizes of 50 or 100 and were considered to have converged for CIFAR10 and SVHN if they reached $100\%$ accuracy and less than 0.0005 loss on the training set. For BHP convergence was considered to have been achieved after 100000 training steps. For all models except VGG16 in Figures 3 and 4 we use learning rates in

$$\big\{0.0075, 0.02, 0.025, 0.03, 0.04, 0.06, 0.07, 0.075, 0.08, 0.09, 0.1, 0.11, 0.12,$$
$$0.13, 0.14, 0.15, 0.16, 0.17, 0.18, 0.19, 0.194, 0.2, 0.22, 0.24, 0.25, 0.26\big\}.$$

VGG16 models were trained with learning rates in $\{0.0075, 0.02, 0.03, 0.06, 0.07, 0.08\}$.

**Network Architectures:**    BHP FCN had 2 hidden layers and were 10 neurons wide. Similarly CIFAR10 FCN were 5 and 7 layers deep with 2048 neurons per layer. 9-layer CONV networks were VGG11 networks with the final 2 layers removed. 16-layer CONV networks were simply the standard implementation of VGG16 networks.

**Run-time:**    The full battery of fully connected models split over two *GeForce GTX 1080* GPUs took two days to train to convergence and the subsequent power iterations took less than a day. Similarly the full gamut of VGG11 models took a day to train to convergence over four *GeForce GTX 1080* GPUs and the subsequent power iterations took less than a day to converge. The VGG16 models took a day to train over four *GeForce GTX 1080* GPUs but the power iterations **for each model** took roughly 24 hours on a single *GeForce GTX 1080* GPU.

## S8    Postponed Proofs

### S8.1    Proof of Proposition 1

*Proof.* Denote $\alpha := \overline{\dim}_H \mu_{W|\mathbf{S}_n}$. By Assumption **H**1, we have

$$\underline{\dim}_{\text{loc}}\mu_{W|\mathbf{S}_n}(w) = \overline{\dim}_{\text{loc}}\mu_{W|\mathbf{S}_n}(w),$$

for $\mu_{W|\mathbf{S}_n}$-a.e. $w$, and by Proposition S6 we have

$$\overline{\dim}_{\text{loc}}\mu_{W|\mathbf{S}_n}(w) \leq \alpha + \epsilon, \tag{S23}$$

for all $\epsilon > 0$ and for $\mu_{W|\mathbf{S}_n}$-a.e. $w$. By invoking Theorem S3, we obtain

$$\overline{\dim}_M \mu_{W|\mathbf{S}_n} \leq \alpha + \epsilon. \tag{S24}$$

Since this holds for any $\epsilon$, $\overline{\dim}_M \mu_{W|\mathbf{S}_n} \leq \alpha$. By definition, we have for almost all $\mathbf{S}_n$:

$$\overline{\dim}_M \mu_{W|\mathbf{S}_n} = \lim_{\delta \to 0} \inf \left\{ \overline{\dim}_M A : \mu_{W|\mathbf{S}_n}(A) \geq 1 - \delta \right\}. \tag{S25}$$

Hence, given a sequence $(\delta_k)_{k \geq 1}$ such that $\delta_k \downarrow 0$, and $\mathbf{S}_n$, and any $\epsilon > 0$, there is a $k_0 = k_0(\epsilon)$ such that $k \geq k_0$ implies

$$\inf \left\{ \overline{\dim}_M A : \mu_{W|\mathbf{S}_n}(A) \geq 1 - \delta_k \right\} \leq \overline{\dim}_M \mu_{W|\mathbf{S}_n} + \epsilon \tag{S26}$$

$$\leq \alpha + \epsilon. \tag{S27}$$

Hence, for any $\epsilon_1 > 0$ and $k \geq k_0$, we can find a bounded Borel set $A_{\mathbf{S}_n,k}$, such that $\mu_{W|\mathbf{S}_n}(A_{\mathbf{S}_n,k}) \geq 1 - \delta_k$, and

$$\overline{\dim}_M A_{\mathbf{S}_n,k} \leq \alpha + \epsilon + \epsilon_1. \tag{S28}$$

Note that the boundedness of $A_{\mathbf{S}_n,k}$ follows from the fact that its upper-Minkowski dimension is finite. By choosing $\epsilon = \epsilon_1 = \frac{\varepsilon}{2}$, it yields the desired result. This completes the proof. $\qquad\square$

## S8.2   Proof of Theorem 1

*Proof.* We begin similarly to the proof of Proposition 1. Denote

$$\alpha(\mathbf{S}, n) := \overline{\dim}_H \mu_{W|\mathbf{S}_n}.$$

By Assumption **H**1, we have $\underline{\dim}_{\mathrm{loc}} \mu_{W|\mathbf{S}_n}(w) = \overline{\dim}_{\mathrm{loc}} \mu_{W|\mathbf{S}_n}(w)$ for $\mu_{W|\mathbf{S}_n}$-almost every $w$, and by Proposition S6 we have

$$\overline{\dim}_{\mathrm{loc}} \mu_{W|\mathbf{S}_n}(w) \leq \alpha(\mathbf{S}, n) + \epsilon, \tag{S29}$$

for all $\epsilon > 0$ and for $\mu_{W|\mathbf{S}_n}$-a.e. $w$. By invoking Theorem S3, we obtain

$$\overline{\dim}_M \mu_{W|\mathbf{S}_n} \leq \alpha(\mathbf{S}, n) + \epsilon. \tag{S30}$$

Since this holds for any $\epsilon > 0$, $\overline{\dim}_M \mu_{W|\mathbf{S}_n} \leq \alpha(\mathbf{S}, n)$. By definition, we have for all $\mathbf{S}$ and $n$:

$$\overline{\dim}_M \mu_{W|\mathbf{S}_n} = \lim_{\delta \to 0} \inf \left\{ \overline{\dim}_M A : \mu_{W|\mathbf{S}_n}(A) \geq 1 - \delta \right\}. \tag{S31}$$

Hence, for a countable sequence of $\delta \downarrow 0$ and each $n$, there exists a set $\Omega_n$ of full measure such that

$$f_\delta^n(\mathbf{S}) := \inf \left\{ \overline{\dim}_M A : \mu_{W|\mathbf{S}_n}(A) \geq 1 - \delta \right\} \to \overline{\dim}_M \mu_{W|\mathbf{S}_n}, \tag{S32}$$

for all $\mathbf{S} \in \Omega_n$. Let $\Omega^* := \cap_n \Omega_n$. Then for $\mathbf{S} \in \Omega^*$ we have that for all $n$

$$f_\delta^n(\mathbf{S}) \to \overline{\dim}_M \mu_{W|\mathbf{S}_n}, \tag{S33}$$

and therefore, on this set we also have

$$\sup_n \frac{1}{\xi_n} \min \left\{ 1, \left| f_\delta^n(\mathbf{S}) - \overline{\dim}_M \mu_{W|\mathbf{S}_n} \right| \right\} \to 0,$$

where $\xi_n$ is a monotone increasing sequence such that $\xi_n \geq 1$ and $\xi_n \to \infty$.

By applying Theorem S4 to the collection of random variables:

$$F_\delta(\mathbf{S}) := \sup_n \frac{1}{\xi_n} \min \left\{ 1, \left| f_\delta^n(\mathbf{S}) - \overline{\dim}_M \mu_{W|\mathbf{S}_n} \right| \right\}, \tag{S34}$$

for any $\zeta > 0$, we can find a subset $\mathfrak{Z} \subset \mathcal{Z}^\infty$, with probability at least $1 - \zeta$ under $\pi^\infty$, such that on $\mathfrak{Z}$ the convergence is uniform, that is

$$\sup_{\mathbf{S} \in \mathfrak{Z}} \sup_n \frac{1}{\xi_n} \min \left\{ 1, \left| f_\delta^n(\mathbf{S}) - \overline{\dim}_M \mu_{W|\mathbf{S}_n} \right| \right\} \leq c(\delta), \tag{S35}$$

where for any $\zeta$, $c(\delta) := c(\delta; \zeta) \to 0$ as $\delta \to 0$. Hence, for any $\delta$, $\mathbf{S} \in \mathfrak{Z}$, and $n$, we have

$$f_\delta^n(\mathbf{S}) \leq \overline{\dim}_{\mathrm{M}} \mu_{W|\mathbf{S}_n} + \xi_n c(\delta) \tag{S36}$$

$$\leq \alpha(\mathbf{S}, n) + \xi_n c(\delta). \tag{S37}$$

Consider a sequence $(\delta_k)_{k \geq 1}$ such that $\delta_k \downarrow 0$ and $\delta_k \in \mathbb{Q}_{>0}$. Then, for any $\mathbf{S} \in \mathfrak{Z}$ and $\epsilon > 0$, we can find a bounded Borel set $A_{\mathbf{S}_n,k}$, such that $\mu_{W|\mathbf{S}_n}(A_{\mathbf{S}_n,k}) \geq 1 - \delta_k$, and

$$\overline{\dim}_{\mathrm{M}} A_{\mathbf{S}_n,k} \leq \alpha(\mathbf{S}, n) + \xi_n c(\delta_k) + \epsilon. \tag{S38}$$

Define the set

$$\mathcal{W}_{n,\delta_k} := \bigcup_{\mathbf{S} \in \mathcal{Z}^\infty} A_{\mathbf{S}_n,k}. \tag{S39}$$

By using $\mathcal{G}(w) := |\mathcal{R}(w) - \hat{\mathcal{R}}(w, \mathbf{S}_n)|$, under the joint distribution of $(W, \mathbf{S}_n)$, such that $\mathbf{S} \sim \pi^\infty$ and $W \sim \mu_{W|\mathbf{S}_n}$, we have:

$$\mathbb{P}\left(\mathcal{G}(W) > \varepsilon\right) \leq \zeta + \mathbb{P}\left(\{\mathcal{G}(W) > \varepsilon\} \cap \{\mathbf{S} \in \mathfrak{Z}\}\right) \tag{S40}$$

$$\leq \zeta + \delta_k + \mathbb{P}\left(\{\mathcal{G}(W) > \varepsilon\} \cap \{W \in A_{\mathbf{S}_n,k}\} \cap \{\mathbf{S} \in \mathfrak{Z}\}\right) \tag{S41}$$

$$\leq \zeta + \delta_k + \mathbb{P}\left(\left\{\sup_{w \in A_{\mathbf{S}_n,k}} \mathcal{G}(w) > \varepsilon\right\} \cap \{\mathbf{S} \in \mathfrak{Z}\}\right). \tag{S42}$$

Now, let us focus on the last term of the above equation. First we observe that as $\ell$ is $L$-Lipschitz, so are $\mathcal{R}$ and $\hat{\mathcal{R}}$. Hence, by considering the particular forms of the $\beta$-covers in **H2**, for any $w' \in \mathbb{R}^d$ we have:

$$\mathcal{G}(w) \leq \mathcal{G}(w') + 2L \|w - w'\|, \tag{S43}$$

which implies

$$\sup_{w \in A_{\mathbf{S}_n,k}} \mathcal{G}(w) \leq \max_{w \in N_{\beta_n}(A_{\mathbf{S}_n,k})} \mathcal{G}(w) + 2L\beta_n. \tag{S44}$$

Now, notice that the $\beta$-covers of **H2** still yield the same Minkowski dimension in (5) [ŞSDE20]. Then by definition, we have for all $\mathbf{S}$ and $n$:

$$\limsup_{\beta \to 0} \frac{\log |N_\beta(A_{\mathbf{S}_n,k})|}{\log(1/\beta)} = \limsup_{\beta \to 0} \sup_{r < \beta} \frac{\log |N_r(A_{\mathbf{S}_n,k})|}{\log(1/r)} = \overline{\dim}_{\mathrm{M}} A_{\mathbf{S}_n,k} := d_{\mathrm{M}}(\mathbf{S}, n, k). \tag{S45}$$

Hence for each $n$

$$g_\beta^{n,k}(\mathbf{S}) := \sup_{\mathbb{Q} \ni r < \delta} \frac{\log |N_r(A_{\mathbf{S}_n,k})|}{\log(1/r)} \to d_{\mathrm{M}}(\mathbf{S}, n, k), \tag{S46}$$

almost surely. By using the same reasoning in (S32), we have, for each $n$, there exists a set $\Omega_n'$ of full measure such that

$$g_\beta^{n,k}(\mathbf{S}) = \sup_{\mathbb{Q} \ni r < \beta} \frac{\log |N_r(A_{\mathbf{S}_n,k})|}{\log(1/r)} \to d_{\mathrm{M}}(\mathbf{S}, n, k), \tag{S47}$$

for all $\mathbf{S} \in \Omega_n'$. Define $\Omega^{**} := (\cap_n \Omega_n') \cap \Omega^*$. Hence, on $\Omega^{**}$ we have:

$$G_\beta^k(\mathbf{S}) := \sup_n \frac{1}{\xi_n} \min\left\{1, \left|g_\beta^{n,k}(\mathbf{S}) - d_{\mathrm{M}}(\mathbf{S}, n, k)\right|\right\} \to 0, \tag{S48}$$

By applying Theorem S4 to the collection $\{G_\beta^k(\mathbf{S})\}_\beta$, for any $\zeta_1 > 0$ we can find a subset $\mathfrak{Z}_1 \subset \mathcal{Z}^\infty$, with probability at least $1 - \zeta_1$ under $\pi^\infty$, such that on $\mathfrak{Z}_1$ the convergence is uniform, that is

$$\sup_{\mathbf{S} \in \mathfrak{Z}_1} \sup_n \frac{1}{\xi_n} \min\{1, |g_\beta^{n,k}(\mathbf{S}) - d_{\mathrm{M}}(\mathbf{S}, n, k)|\} \leq c'(\beta), \tag{S49}$$

where for any $\zeta_1$, $c'(\beta) := c'(\beta; \zeta_1, \delta_k) \to 0$ as $\beta \to 0$. Hence, denoting $\mathfrak{Z}^* := \mathfrak{Z} \cap \mathfrak{Z}_1$ by using (S38) we have:

$$\{\mathbf{S} \in \mathfrak{Z}^*\} \subseteq \bigcap_n \left\{|N_\beta(A_{\mathbf{S}_n,k})| \leq \left(\frac{1}{\beta}\right)^{\alpha(\mathbf{S},n) + \xi_n c(\delta_k) + \xi_n c'(\beta) + \epsilon}\right\}.$$

Let $(\beta_n)_{n \geq 0}$ be a decreasing sequence such that $\beta_n \in \mathbb{Q}$ for all $n$ and $\beta_n \to 0$. We then have

$$\mathbb{P}\left( \{\mathbf{S} \in \mathfrak{Z}\} \cap \left\{ \max_{w \in N_{\beta_n}(A_{\mathbf{S}_n,k})} \mathcal{G}_n(w) \geq \varepsilon \right\} \right)$$

$$\leq \mathbb{P}\left( \{\mathbf{S} \in \mathfrak{Z}^*\} \cap \left\{ \max_{w \in N_{\beta_n}(A_{\mathbf{S}_n,k})} \mathcal{G}_n(w) \geq \varepsilon \right\} \right) + \zeta_2.$$

For $\rho > 0$ and $m \in \mathbb{N}_+$ let us define the interval $J_m(\rho) := (m\rho, (m+1)\rho]$. Furthermore, for any $t > 0$ define

$$\varepsilon(t) := \sqrt{\frac{2\nu^2}{n}\left[ \log(1/\beta_n)\left(t + \xi_n c(\delta_k) + \xi_n c'(\beta_n) + \epsilon\right) + \log(M/\zeta_2) \right]}. \tag{S50}$$

For notational simplicity, denote $N_{\beta_n,k} := N_{\beta_n}(\mathcal{W}_{n,\delta_k})$, where $\mathcal{W}_{n,\delta_k}$ is defined in (S39) and

$$\tilde{\alpha}(\mathbf{S}, n, k, \epsilon) := \alpha(\mathbf{S}, n) + \xi_n c(\delta_k) + \xi_n c'(\beta_n) + \epsilon. \tag{S51}$$

Let $d^*$ be the smallest real number such that $\alpha(\mathbf{S}, n) \leq d^*$ almost surely[4], we therefore have:

$$\mathbb{P}\left( \{\mathbf{S} \in \mathfrak{Z}\} \cap \left\{ \max_{w \in N_{\beta_n}(A_{\mathbf{S}_n,k})} \mathcal{G}_n(w) \geq \varepsilon(\alpha(\mathbf{S}, n)) \right\} \right)$$

$$\leq \zeta_2 + \mathbb{P}\left( \left\{ |N_{\beta_n}(A_{\mathbf{S}_n,k})| \leq \left(\frac{1}{\beta_n}\right)^{\tilde{\alpha}(\mathbf{S},n,k,\epsilon)} \right\} \right.$$

$$\left. \cap \left\{ \max_{w \in N_{\beta_n}(A_{\mathbf{S}_n,k})} |\hat{\mathcal{R}}_n(w) - \mathcal{R}(w)| \geq \varepsilon(\alpha(\mathbf{S}, n)) \right\} \right)$$

$$= \zeta_2 + \sum_{m=0}^{\lceil \frac{d^*}{\rho} \rceil} \mathbb{P}\left( \left\{ |N_{\beta_n}(A_{\mathbf{S}_n,k})| \leq \left(\frac{1}{\beta_n}\right)^{\tilde{\alpha}(\mathbf{S},n,k,\epsilon)} \right\} \right.$$

$$\left. \cap \left\{ \max_{w \in N_{\beta_n}(A_{\mathbf{S}_n,k})} \mathcal{G}_n(w) \geq \varepsilon(\alpha(\mathbf{S}, n)) \right\} \cap \{\alpha(\mathbf{S}, n) \in J_m(\rho)\} \right)$$

$$= \zeta_2 + \sum_{m=0}^{\lceil \frac{d^*}{\rho} \rceil} \mathbb{P}\left( \left\{ |N_{\beta_n}(A_{\mathbf{S}_n,k})| \leq \left(\frac{1}{\beta_n}\right)^{\tilde{\alpha}(\mathbf{S},n,k,\epsilon)} \right\} \cap \{\alpha(\mathbf{S}, n) \in J_m(\rho)\} \right.$$

$$\left. \cap \bigcup_{w \in N(\beta_n)} \left( \{w \in N_{\beta_n}(A_{\mathbf{S}_n,k})\} \cap \{\mathcal{G}_n(w) \geq \varepsilon(\alpha(\mathbf{S}, n))\} \right) \right)$$

$$\leq \zeta_2 + \sum_{m=0}^{\lceil \frac{d^*}{\rho} \rceil} \sum_{w \in N_{\beta_n,k}} \mathbb{P}\left( \{\mathcal{G}_n(w) \geq \varepsilon(m\rho)\} \right.$$

$$\left. \cap \left\{ w \in N_{\delta_n}(A_{\mathbf{S}_n,k}) \right\} \cap \left\{ |N_{\beta_n}(A_{\mathbf{S}_n,k})| \leq \left(\frac{1}{\beta_n}\right)^{\tilde{\alpha}(\mathbf{S},n,k,\epsilon)} \right\} \cap \{\alpha(\mathbf{S}, n) \in J_m(\rho)\} \right),$$

where we used the fact that on the event $\alpha(\mathbf{S}, n) \in J_m(\rho)$, $\varepsilon(\alpha(\mathbf{S}, n)) \geq \varepsilon(m\rho)$.

Notice that the events

$$\left\{ w \in N_{\beta_n}(A_{\mathbf{S}_n,k}) \right\}, \left\{ |N_{\beta_n}(A_{\mathbf{S}_n,k})| \leq (1/\beta_n)^{\tilde{\alpha}(\mathbf{S},n,k,\epsilon)} \right\}, \{\alpha(\mathbf{S}, n) \in J_m(\rho)\}$$

are in $\mathfrak{G}$. On the other hand, the event $\{\mathcal{G}_n(w) \geq \varepsilon(m\rho)\}$ is clearly in $\mathfrak{F}$ (see **H**2 for definitions).

---

[4]Notice that we trivially have $d^* \leq d$; yet, $d^*$ can be much smaller than $d$.

Therefore, we have

$$\mathbb{P}\left( \{ \mathbf{S} \in \mathfrak{Z} \} \cap \left\{ \max_{w \in N_{\beta_n}(A_{\mathbf{S}_n,k})} \mathcal{G}_n(w) \geq \varepsilon(\alpha(\mathbf{S},n)) \right\} \right)$$

$$\leq \zeta_2 + M \sum_{m=0}^{\lceil \frac{d^*}{\rho} \rceil} \sum_{w \in N_{\beta_n,k}} \mathbb{P}\left( \left\{ \mathcal{G}_n(w) \geq \varepsilon(m\rho) \right\} \right)$$

$$\times \mathbb{P}\left( \left\{ w \in N_{\beta_n}(A_{\mathbf{S}_n,k}) \right\} \cap \left\{ |N_{\beta_n}(A_{\mathbf{S}_n,k})| \leq \left( \frac{1}{\beta_n} \right)^{\tilde{\alpha}(\mathbf{S},n,k,\epsilon)} \right\} \cap \{ \alpha(\mathbf{S},n) \in J_m(\rho) \} \right),$$

Recall that $\mathcal{G}_n(w) = \frac{1}{n} \sum_{i=1}^n [\ell(w, z_i) - \mathbb{E}_{z \sim \pi} \ell(w, z)]$. Since the $(z_i)_i$ are i.i.d. by Assumption 3 it follows that $\mathcal{G}_n(w)$ is $(\nu/\sqrt{n}, \kappa/n)$-sub-exponential and from [Wai19, Proposition 2.9] we have that

$$\mathbb{P}\left( \left\{ \mathcal{G}_n(w) \geq \varepsilon(m\rho) \right\} \right) \leq 2 \exp\left( -\frac{n\varepsilon(m\rho)^2}{2\nu^2} \right),$$

as long as $\varepsilon(m\rho) \leq \nu^2/\kappa$. For $n$ large enough we may assume that $\varepsilon(d^*) \leq \nu^2/\kappa$, and thus

$$\mathbb{P}\left( \{ \mathbf{S} \in \mathfrak{Z} \} \cap \left\{ \max_{w \in N_{\beta_n}(A_{\mathbf{S}_n,k})} \mathcal{G}_n(w) \geq \varepsilon(\alpha(\mathbf{S},n)) \right\} \right)$$

$$\leq 2M \sum_{m=0}^{\lceil \frac{d}{\rho} \rceil} e^{-\frac{2n\varepsilon^2(m\rho)}{B^2}} \sum_{w \in N_{\beta_n,k}} \mathbb{P}\left( \left\{ w \in N_{\beta_n}(A_{\mathbf{S}_n,k}) \right\} \cap \left\{ |N_{\beta_n}(A_{\mathbf{S}_n,k})| \leq \left( \frac{1}{\beta_n} \right)^{\tilde{\alpha}(\mathbf{S},n,k,\epsilon)} \right\} \right.$$

$$\left. \cap \{ \alpha(\mathbf{S},n) \in J_m(\rho) \} \right) + \zeta_2$$

$$\leq 2M \sum_{m=0}^{\lceil \frac{d}{\rho} \rceil} e^{-\frac{n\varepsilon^2(m\rho)}{2\nu^2}} \sum_{w \in N_{\beta_n,k}} \mathbb{E}\left[ \mathbb{1}\left\{ w \in N_{\beta_n}(A_{\mathbf{S}_n,k}) \right\} \right.$$

$$\left. \times \mathbb{1}\left\{ |N_{\beta_n}(A_{\mathbf{S}_n,k})| \leq \left( \frac{1}{\beta_n} \right)^{\tilde{\alpha}(\mathbf{S},n,k,\epsilon)} \right\} \times \mathbb{1}\left\{ \alpha(\mathbf{S},n) \in J_m(\rho) \right\} \right] + \zeta_2$$

$$\leq 2M \sum_{m=0}^{\lceil \frac{d}{\rho} \rceil} e^{-\frac{n\varepsilon^2(m\rho)}{2\nu^2}} \mathbb{E}\left[ \sum_{w \in N_{\beta_n,k}} \mathbb{1}\left\{ w \in N_{\beta_n}(A_{\mathbf{S}_n,k}) \right\} \right.$$

$$\left. \times \mathbb{1}\left\{ |N_{\beta_n}(A_{\mathbf{S}_n,k})| \leq \left( \frac{1}{\beta_n} \right)^{\tilde{\alpha}(\mathbf{S},n,k,\epsilon)} \right\} \times \mathbb{1}\left\{ \alpha(\mathbf{S},n) \in J_m(\rho) \right\} \right] + \zeta_2 \quad \text{(S52)}$$

$$\leq 2M \sum_{m=0}^{\lceil \frac{d}{\rho} \rceil} e^{-\frac{n\varepsilon^2(m\rho)}{2\nu^2}} \mathbb{E}\left[ |N_{\beta_n}(A_{\mathbf{S}_n,k})| \times \mathbb{1}\left\{ |N_{\beta_n}(A_{\mathbf{S}_n,k})| \leq \left( \frac{1}{\beta_n} \right)^{\tilde{\alpha}(\mathbf{S},n,k,\epsilon)} \right\} \right.$$

$$\left. \times \mathbb{1}\left\{ \alpha(\mathbf{S},n) \in J_m(\rho) \right\} \right] + \zeta_2$$

$$= \zeta_2 + 2M \sum_{m=0}^{\lceil \frac{d}{\rho} \rceil} e^{-\frac{n\varepsilon^2(m\rho)}{2\nu^2}} \mathbb{E}\left[ \left[ \frac{1}{\beta_n} \right]^{\tilde{\alpha}(\mathbf{S},n,k,\epsilon)} \times \mathbb{1}\left\{ \alpha(\mathbf{S},n) \in J_m(\rho) \right\} \right],$$

where (S52) follows from Fubini's theorem.

Now, notice that the mapping $t \mapsto \varepsilon^2(t)$ is linear with derivative bounded by

$$\frac{2\nu^2}{n} \log(1/\beta_n).$$

Therefore, on the event $\{\alpha(\mathbf{S}, n) \in J_m(\rho)\}$ we have

$$\varepsilon^2(\alpha(\mathbf{S}, n)) - \varepsilon^2(m\rho) \leq (\alpha(\mathbf{S}, n) - m\rho)\frac{2\nu^2}{n}\log(1/\beta_n) \tag{S53}$$

$$\leq \rho\frac{2\nu^2}{n}\log(1/\beta_n). \tag{S54}$$

By choosing $\rho = \rho_n = 1/\log(1/\beta_n)$, we have

$$\varepsilon^2(m\rho_n) \geq \varepsilon^2(\alpha(\mathbf{S}, n)) - \frac{2\nu^2}{n}.$$

Therefore, we have

$$\mathbb{P}\left( \{\mathbf{S} \in \mathfrak{Z}\} \cap \left\{ \max_{w \in N_{\beta_n}(A_{\mathbf{S}_n, k})} \mathcal{G}_n(w) \geq \varepsilon(\alpha(\mathbf{S}, n)) \right\} \right)$$

$$\leq \zeta_2 + 2M\mathbb{E}\left[ \sum_{m=0}^{\lceil \frac{d}{\rho_n} \rceil} e^{-\frac{n\varepsilon^2(m\rho_n)}{2\nu^2}} \left[ \frac{1}{\beta_n} \right]^{\tilde{\alpha}(\mathbf{S}, n, k, \epsilon)} \times \mathbb{1}\{\alpha(\mathbf{S}, n) \in J_m(\rho_n)\} \right]$$

$$\leq \zeta_2 + 2M\mathbb{E}\left[ \sum_{m=0}^{\lceil \frac{d}{\rho_n} \rceil} e^{-\frac{n\varepsilon^2(\alpha(\mathbf{S}, n))}{2\nu^2}+1} \left[ \frac{1}{\beta_n} \right]^{\tilde{\alpha}(\mathbf{S}, n, k, \epsilon)} \times \mathbb{1}\{\alpha(\mathbf{S}, n) \in J_m(\rho_n)\} \right]$$

$$= \zeta_2 + 2M\mathbb{E}\left[ e^{-\frac{n\varepsilon^2(\alpha(\mathbf{S}, n))}{2\nu^2}+1} \left[ \frac{1}{\beta_n} \right]^{\tilde{\alpha}(\mathbf{S}, n, k, \epsilon)} \right].$$

By the definition of $\varepsilon(t)$, for any $\mathbf{S}$ and $n$ we have that:

$$2Me^{-\frac{n\varepsilon^2(\alpha(\mathbf{S}, n))}{2\nu^2}+1} \left[ \frac{1}{\beta_n} \right]^{\tilde{\alpha}(\mathbf{S}, n, k, \epsilon)} = 2e\zeta_2.$$

Therefore,

$$\mathbb{P}\left( \{\mathbf{S} \in \mathfrak{Z}\} \cap \left\{ \max_{w \in N_{\beta_n}(A_{\mathbf{S}_n, k})} \mathcal{G}_n(w) \geq \varepsilon(\alpha(\mathbf{S}, n)) \right\} \right) \leq (1 + 2e)\zeta_2.$$

Therefore, by using the definition of $\varepsilon(t)$, (S42), and (S44), with probability at least $1 - \zeta - \delta_k - (1 + 2e)\zeta_2$, we have

$$|\hat{\mathcal{R}}(W, \mathbf{S}_n) - \mathcal{R}(W)| \leq 2\sqrt{\frac{2\nu^2}{n}\left[ \log\left(\frac{1}{\beta_n}\right)\left(\alpha(\mathbf{S}, n) + \xi_n c(\delta_k) + \xi_n c'(\beta_n) + \epsilon\right) + \log\left(\frac{M}{\zeta_2}\right) \right]}$$
$$+ 2L\beta_n.$$

Choose $k$ such that $\delta_k \leq \zeta/2$, $\zeta_2 = \zeta/(2+4e)$, $\xi_n = \log\log(n)$, $\epsilon = \alpha(\mathbf{S}, n)$, and $\beta_n = \sqrt{2\nu^2/L^2 n}$. Then, with probability at least $1 - 2\zeta$, we have

$$|\hat{\mathcal{R}}(W, \mathbf{S}_n) - \mathcal{R}(W)| \tag{S55}$$

$$\leq 4\sqrt{\frac{4\nu^2}{n}\left[ \frac{1}{2}\log\left(nL^2\right)\left(2\alpha(\mathbf{S}, n) + c(\delta_k)\log\log(n) + o(\log\log(n))\right) + \log\left(\frac{13M}{\zeta}\right) \right]}. \tag{S56}$$

Finally, as we have $\alpha(\mathbf{S}, n)\log(n) = \omega(\log\log(n))$, for $n$ large enough, we obtain

$$|\hat{\mathcal{R}}(W, \mathbf{S}_n) - \mathcal{R}(W)| \leq 8\nu\sqrt{\frac{\alpha(\mathbf{S}, n)\log^2\left(nL^2\right)}{n} + \frac{\log\left(13M/\zeta\right)}{n}}. \tag{S57}$$

This completes the proof. $\qquad\square$

## S8.3 Proof of Proposition 2

*Proof.* If we apply SGD this results in the recursion (6) with

$$h_i(w) = M_i w + q_i \quad \text{with} \quad M_i := (1 - \eta\lambda)I - \eta H_i, \tag{S58}$$

$$H_i := \frac{1}{b} \sum_{j \in S_i} a_j a_j^T, \quad q_i := (\eta/b) \sum_{j \in S_i} a_j y_j,$$

where $a_j \in \mathbb{R}^d$ are the input vector, and $y_j$ are the output variable, and $\{S_i\}_{i=1}^{m_b}$ is a partition of $\{1, 2, \ldots, n\}$ with $|S_i| = b$ with $i = 1, 2, \ldots, m_b$ and $m_b = n/b$. Let $L_i$ be the Lipschitz constant of $\nabla\ell(w, z_i)$. It can be seen that $\nabla\ell(w, z_i)$ is Lipschitz with constant $L_i = R_i^2 + \lambda$, where $R_i = \max_{j \in S_i} \|a_j\|$. We assume $\eta < 2/L = 2/(R^2 + \lambda)$, where $R = \max_i R_i$, otherwise the expectation of the iterates can diverge from some initializations and for some choices of the batch-size. We have

$$h_i(u) - h_i(v) = M_i(u - v),$$

where

$$0 \preceq \left(1 - \eta\lambda - \eta R_i^2\right) I \preceq M_i \preceq (1 - \eta\lambda)I.$$

Hence, $h_i$ is bi-Lipschitz in the sense of [Anc16] where

$$\gamma_i \|u - v\| \le \|h_i(u) - h_i(v)\| \le \Gamma_i \|u - v\|,$$

with

$$\gamma_i = \min\left(\left|1 - \eta\lambda - \eta R_i^2\right|, |1 - \eta\lambda|\right), \tag{S59}$$

$$\Gamma_i = \max\left(\left|1 - \eta\lambda - \eta R_i^2\right|, |1 - \eta\lambda|\right) < 1, \tag{S60}$$

as long as $\gamma_i > 0$. For simplicity of the presentation, we assume $\eta < \frac{1}{R^2 + \lambda}$ in which case the expressions for $\gamma_i$ and $\Gamma_i$ simplify to:

$$\gamma_i = 1 - \eta\lambda - \eta R_i^2, \quad \Gamma_i = 1 - \eta\lambda.$$

In this case, it is easy to see that

$$0 < \gamma_i \le \|J_{h_i}(w)\| \le \Gamma_i < 1,$$

and it follows from Theorem S2 that

$$\overline{\dim}_{\mathrm{H}} \mu_{W|\mathbf{S}_n} \le \frac{\mathcal{E}}{\sum_{i=1}^{m_b} p_i \log(\Gamma_i)} = \frac{-\mathcal{E}}{\sum_{i=1}^{m_b} p_i \log(1/\Gamma_i)}. \tag{S61}$$

By Jensen's inequality, we have

$$-\mathcal{E} = \sum_{i=1}^{m_b} p_i \log\left(\frac{1}{p_i}\right) \le \log\left(\sum_{i=1}^{m_b} p_i \cdot \frac{1}{p_i}\right) = \log(m_b), \tag{S62}$$

where $m_b = n/b$. When $\eta < \frac{1}{R^2 + \lambda}$, we recall that $\gamma_i = 1 - \eta\lambda - \eta R_i^2$ and $\Gamma_i = 1 - \eta\lambda$. Therefore,

$$\overline{\dim}_{\mathrm{H}} \mu_{W|\mathbf{S}_n} \le \frac{-\mathcal{E}}{\sum_{i=1}^{m_b} p_i \log(1/\Gamma_i)} \le \frac{\log(m_b)}{\log(1/(1 - \eta\lambda))} = \frac{\log(n/b)}{\log(1/(1 - \eta\lambda))}. \tag{S63}$$

The proof is complete. $\qquad\square$

## S8.4 Proof of Proposition 3

*Proof.* When the batch-size is equal to $b$, we can compute that the Jacobian is given by

$$J_{h_i}(w) = \frac{1}{b} \sum_{j \in S_i} \left(1 - \eta\lambda + \eta y_j^2 \left[\frac{e^{-y_j a_j^T w}}{(1 + e^{-y_j a_j^T w})^2}\right] a_j a_j^T\right), \tag{S64}$$

where $\{S_i\}_{i=1}^{m_b}$ is a partition of $\{1, 2, \ldots, n\}$ with $|S_i| = b$, where $i = 1, 2, \ldots, m_b$ and $m_b = n/b$. Note that the input data is bounded, i.e. $R_i := \max_{j \in S_i} \|a_j\| < \infty$, and $R := \max_i R_i < 2\sqrt{\lambda}$.

Recall that the step-size is sufficiently small, i.e. $\eta < 1/\lambda$. One can provide the upper bound on $J_{h_i}(w)$:

$$\|J_{h_i}(w)\| \leq \Gamma_i := 1 - \eta\lambda + \frac{1}{4}\eta R_i^2 \leq 1 - \eta\lambda + \frac{1}{4}\eta R^2, \tag{S65}$$

so that

$$\overline{\dim}_H \mu_{W|\mathbf{S}_n} \leq \frac{\mathcal{E}}{\sum_{i=1}^{m_b} p_i \log(\Gamma_i)} = \frac{-\mathcal{E}}{\sum_{i=1}^{m_b} p_i \log(1/(1 - \eta\lambda + \frac{1}{4}\eta R_i^2))}$$

$$\leq \frac{\log m_b}{\log(1/(1 - \eta\lambda + \frac{1}{4}\eta R^2))} \tag{S66}$$

$$= \frac{\log(n/b)}{\log(1/(1 - \eta\lambda + \frac{1}{4}\eta R^2))}, \tag{S67}$$

where we used (S65) and (S62) in (S66). The proof is complete. $\qquad\square$

## S8.5  Proof of Proposition 4

*Proof.* We can compute that

$$\nabla\ell(w, z_i) = -a_i\rho'(y_i - \langle w, a_i\rangle) + \lambda w, \tag{S68}$$

$$h_i(w) = \frac{1}{b}\sum_{j\in S_i}(1 - \eta\lambda)w + \eta a_j\rho'(y_j - \langle w, a_j\rangle), \tag{S69}$$

$$J_{h_i}(w) = \frac{1}{b}\sum_{j\in S_i}(1 - \eta\lambda)I - \eta a_j a_j^T \rho''(y_j - \langle w, a_j\rangle), \tag{S70}$$

where $\{S_i\}_{i=1}^{m_b}$ is a partition of $\{1, 2, \ldots, n\}$ with $|S_i| = b$, where $i = 1, 2, \ldots, m_b$ with $m_b = n/b$. Furthermore, $\|\rho''_{\exp}\|_\infty = \rho''_{\exp}(0) = \frac{2}{t_0}$. Therefore, for $\eta \in (0, \frac{1}{\lambda + R^2(2/t_0)})$,

$$0 < (1 - \eta\lambda) - \eta R^2 \frac{2}{t_0} \leq \|J_{h_i}(w)\| \leq (1 - \eta\lambda) + \eta R^2 \frac{2}{t_0}, \tag{S71}$$

where $R = \max_i \|a_i\| < \sqrt{\lambda t_0/2}$. We have

$$\overline{\dim}_H \mu_{W|\mathbf{S}_n} \leq \frac{\log m_b}{\log(1/(1 - \eta\lambda + \eta R^2 \frac{2}{t_0}))} = \frac{\log(n/b)}{\log(1/(1 - \eta\lambda + \eta R^2 \frac{2}{t_0}))}, \tag{S72}$$

where we used (S71) and (S62). The proof is complete. $\qquad\square$

## S8.6  Proof of Proposition S7

*Proof.* We can compute that

$$\nabla\ell(w, z_i) = y_i\ell'_\sigma(y_i a_i^T w)a_i + \lambda w,$$
$$\nabla^2\ell(w, z_i) = y_i^2\ell''_\sigma(y_i a_i^T w)a_i a_i^T + \lambda,$$
$$h_i(w) = w - \frac{\eta}{b}\sum_{j\in S_i}\nabla\ell(w, z_j),$$

where $\{S_i\}_{i=1}^{m_b}$ is a partition of $\{1, 2, \ldots, n\}$ with $|S_i| = b$, where $i = 1, 2, \ldots, m_b$ with $m_b = n/b$, so that

$$J_{h_i}(w) = I - \frac{\eta}{b}\sum_{j\in S_i}\nabla^2\ell(w, z_j) = (1 - \eta\lambda)I - \frac{\eta}{b}\sum_{j\in S_i}y_j^2\ell''_\sigma(y_j a_j^T w)a_j a_j^T,$$

with

$$\ell''_\sigma(z) = \frac{1}{\sigma}\frac{e^{-(1-z)/\sigma}}{(1 + e^{-(1-z)/\sigma})^2} \geq 0, \quad \|\ell''_\sigma\|_\infty = \ell''_\sigma(1) = \frac{1}{4\rho}.$$

Therefore, if $\eta \in (0, \frac{1}{\lambda + \|R\|^2/(4\rho)})$ and $R := \max_i \|a_i\|$, then

$$1 - \eta\lambda - \eta\frac{1}{4\rho}R^2 \leq \|J_{h_i}(w)\| \leq 1 - \eta\lambda.$$

This implies that

$$\overline{\dim}_{\mathrm{H}}\mu_{W|\mathbf{S}_n} \leq \frac{\log m_b}{\log(1/(1-\eta\lambda))} = \frac{\log(n/b)}{\log(1/(1-\eta\lambda))}, \tag{S73}$$

where we used (S62). The proof is complete. $\qquad\square$

### S8.7 Proof of Proposition 5

*Proof.* We recall that the loss is given by:

$$\ell(w, z_i) := \|y_i - \hat{y}_i\|^2 + \lambda\|w\|^2/2, \quad \hat{y}_i := \sum_{r=1}^{m} b_r\sigma\left(w_r^T a_i\right), \tag{S74}$$

where the non-linearity $\sigma : \mathbb{R} \to \mathbb{R}$ is smooth and $\lambda > 0$ is a regularization parameter. Note that we can re-write (S74) as $\ell(w, z_i) = \left\|y_i - b^T\sigma\left(w_r^T a_i\right)\right\|^2 + \lambda\|w\|^2/2$. We can compute that

$$\frac{\partial\ell(w, z_i)}{\partial w_r} = -(y_i - \hat{y}_i)\frac{\partial\hat{y}_i}{\partial w_r} + \lambda w_r = -(y_i - \hat{y}_i)b_r\sigma'(w_r^T a_i)a_i + \lambda w_r. \tag{S75}$$

Therefore,

$$\nabla\ell(w, z_i) = -(y_i - \hat{y}_i)v_i + \lambda w, \quad \text{where} \quad v_i := \begin{bmatrix} b_1\sigma'(w_1^T a_i)a_i \\ b_2\sigma'(w_2^T a_i)a_i \\ \cdots \\ b_m\sigma'(w_m^T a_i)a_i \end{bmatrix},$$

with

$$h_i(w) = w - \frac{\eta}{b}\sum_{j \in S_i}\nabla\ell(w, z_i),$$

and

$$J_{h_i}(w) = (1 - \eta\lambda)I - \frac{\eta}{b}\sum_{j \in S_i} v_j \otimes v_j^T$$

$$+ \frac{\eta}{b}\sum_{j \in S_i}(y_j - \hat{y}_j)\begin{bmatrix} b_1\sigma''(w_1^T a_j)a_j a_j^T & 0_d & \cdots & 0_d \\ 0_d & b_2\sigma''(w_2^T a_j)a_j a_j^T & \cdots & 0_d \\ \vdots & \vdots & \ddots & \vdots \\ 0_d & 0_d & \cdots & b_m\sigma''(w_m^T a_j)a_j a_j^T \end{bmatrix}$$

$$= (1 - \eta\lambda)I - \frac{\eta}{b}\sum_{j \in S_i}\mathrm{diag}(\{B_r^{(j)}\}_{r=1}^m), \tag{S76}$$

where $\{S_i\}_{i=1}^{m_b}$ is a partition of $\{1, 2, \ldots, n\}$ with $|S_i| = b$, where $i = 1, 2, \ldots, m_b$ with $m_b = n/b$, and

$$B_r^{(i)} := b_r\left[-(y_i - \hat{y}_i)\sigma''(w_r^T a_i) + (\sigma'(w_r^T a_i))^2\right]a_i a_i^T, \tag{S77}$$

and $0_d$ is a $d \times d$ zero matrix and $\mathrm{diag}(\{B_r^{(i)}\}_{r=1}^m)$ denotes a block diagonal matrix with the matrices $B_r^{(i)}$ on the diagonal. We assume the output $y_i$ and the activation function $\sigma$ and its second derivative $\sigma''$ is bounded.[5] This would for instance clearly hold for classification problems where $y_i$ can take integer values on a compact set with a sigmoid or hyperbolic tangent activation function. Then, under this assumption, there exists a constant $M_y > 0$ such that $\max_i \|y_i - \hat{y}_i\| \leq M_y$. Then for $\eta \in (0, \frac{1}{2\lambda})$ and $\lambda > C$ where $C := M_y\|b\|_\infty\|\sigma''\|_\infty R^2 + (\max_j \|v_j\|_\infty)^2$, we get

$$1 - \eta(C + \lambda) \leq \|J_{h_i}(w)\| \leq 1 - \eta(\lambda - C).$$

This implies that

$$\overline{\dim}_{\mathrm{H}}\mu_{W|\mathbf{S}_n} \leq \frac{\log m_b}{\log(1/(1-\eta(\lambda-C)))} = \frac{\log(n/b)}{\log(1/(1-\eta(\lambda-C)))}, \tag{S78}$$

where we used (S62). The proof is complete. $\qquad\square$

---

[5] Since the final layer is fixed at initialization, the output is bounded.

## S8.8 Proof of Proposition S8

*Proof.* Recall that $H$ is positive-definite and there exist some $m, M > 0$:

$$0 \prec mI \preceq H \preceq MI. \tag{S79}$$

We have

$$h_i(u) - h_i(v) = M_i(u - v),$$

where

$$0 \preceq \left(1 - \eta\lambda m^{-1} - \eta m^{-1} R_i^2\right) I \preceq M_i \preceq \left(1 - \eta\lambda M^{-1}\right) I, \tag{S80}$$

where $R_i := \max_{j \in S_i} \|a_j\|$, and we recall the assumption that $\eta < \frac{m}{R^2 + \lambda}$, with $R := \max_i R_i$. Hence, $h_i$ is bi-Lipschitz in the sense of [Anc16] where

$$\gamma_i(u - v) \le \|h_i(u) - h_i(v)\| \le \Gamma_i(u - v),$$

with

$$\gamma_i = \min\left(\left|1 - \eta\lambda m^{-1} - \eta m^{-1} R_i^2\right|, \left|1 - \eta M^{-1}\lambda\right|\right), \tag{S81}$$

$$\Gamma_i = \max\left(\left|1 - \eta\lambda m^{-1} - \eta m^{-1} R_i^2\right|, \left|1 - \eta M^{-1}\lambda\right|\right) < 1, \tag{S82}$$

as long as $\gamma_i > 0$. We recall the assumption $\eta < \frac{m}{R^2 + \lambda}$, where $R := \max_i R_i$, in which case the expressions for $\gamma_i$ and $\Gamma_i$ simplify to:

$$\gamma_i = 1 - \eta m^{-1}\lambda - \eta m^{-1} R_i^2, \quad \Gamma_i = 1 - \eta M^{-1}\lambda.$$

In this case, it is easy to see that

$$0 < \gamma_i \le \|J_{h_i}(w)\| \le \Gamma_i < 1,$$

and it follows from Theorem S2 that

$$\overline{\dim}_{\mathrm{H}} \mu_{W|\mathbf{S}_n} \le \frac{\mathcal{E}}{\sum_{i=1}^{m_b} p_i \log(\Gamma_i)} \le \frac{\log m_b}{\log(1/(1 - \eta M^{-1}\lambda))} = \frac{\log(n/b)}{\log(1/(1 - \eta M^{-1}\lambda))}, \tag{S83}$$

where we used (S62). The proof is complete. $\qquad\square$

## S8.9 Proof of Proposition S9

*Proof.* Similar as in the proof of Proposition 3, we can compute that the Jacobian is given by

$$J_{h_i}(w) = \frac{1}{b} \sum_{j \in S_i} \left(1 - \eta H^{-1}\lambda + \eta H^{-1} y_j^2 \left[\frac{e^{-y_j a_j^T w}}{(1 + e^{-y_j a_j^T w})^2}\right] a_j a_j^T\right), \tag{S84}$$

where $\{S_i\}_{i=1}^{m_b}$ is a partition of $\{1, 2, \ldots, n\}$ with $|S_i| = b$, where $i = 1, 2, \ldots, m_b$ with $m_b = n/b$, and $H$ is a positive-definite matrix with $0 \prec mI \preceq H \preceq MI$. recall that the input data is bounded, i.e. $\max_{j \in S_i} \|a_j\| \le R_i$ for some $R_i$, and $R := \max_i R_i$ satisfying $R < 2\sqrt{m\lambda/M}$. Also recall the step-size is sufficiently small, i.e. $\eta < m/\lambda$. One can provide upper bounds and lower bounds on $J_{h_i}(w)$:

$$\|J_{h_i}(w)\| \le \Gamma_i := 1 - \eta M^{-1}\lambda + \frac{1}{4}\eta m^{-1} R_i^2, \tag{S85}$$

$$\|J_{h_i}(w)\| \ge \gamma_i := 1 - \eta m^{-1}\lambda, \tag{S86}$$

so that

$$\overline{\dim}_{\mathrm{H}} \mu_{W|\mathbf{S}_n} \le \frac{-\mathcal{E}}{\sum_{i=1}^{m_b} p_i \log(1/(1 - \eta M^{-1}\lambda + \frac{1}{4}\eta m^{-1} R_i^2))}$$

$$\le \frac{\log m_b}{\log(1/(1 - \eta M^{-1}\lambda + \frac{1}{4}\eta m^{-1} R^2))} \tag{S87}$$

$$= \frac{b \log(n/b)}{\log(1/(1 - \eta M^{-1}\lambda + \frac{1}{4}\eta m^{-1} R^2))}, \tag{S88}$$

where we used (S62) in (S87). The proof is complete. $\qquad\square$

### S8.10 Proof of Proposition S10

*Proof.* Similar as in the proof of Proposition 4, we can compute that

$$J_{h_i}(w) = \frac{1}{b} \sum_{j \in S_i} \left(I - \eta H^{-1}\lambda\right) - \eta H^{-1} a_j a_j^T \rho'' \left(y_j - \langle w, a_j \rangle\right), \tag{S89}$$

where $\{S_i\}_{i=1}^{m_b}$ is a partition of $\{1, 2, \ldots, n\}$ with $|S_i| = b$, where $i = 1, 2, \ldots, m_b$ with $m_b = n/b$, and $H$ is a positive-definite matrix with $0 \prec mI \preceq H \preceq MI$. For the function $\rho$, a standard choice is exponential squared loss: $\rho_{\exp}(t) = 1 - e^{-|t|^2/t_0}$, where $t_0 > 0$ is a tuning parameter. We can compute that $\|\rho''_{\exp}\|_\infty = \rho''_{\exp}(0) = \frac{2}{t_0}$. Therefore, for $\eta \in (0, \frac{m}{\lambda + R^2(2/t_0)})$,

$$0 < 1 - \eta m^{-1}\lambda - \eta m^{-1} R^2 \frac{2}{t_0} \leq \|J_{h_i}(w)\| \leq 1 - \eta M^{-1}\lambda + \eta m^{-1} R^2 \frac{2}{t_0}, \tag{S90}$$

where $R = \max_i \|a_i\|$ and we recall that $R < \sqrt{\lambda t_0 m/(2M)}$. We have

$$\overline{\dim}_H \mu_{W|\mathbf{S}_n} \leq \frac{\log m_b}{\log(1/(1 - \eta M^{-1}\lambda + \eta m^{-1} R^2 \frac{2}{t_0}))} = \frac{\log(n/b)}{\log(1/(1 - \eta M^{-1}\lambda + \eta m^{-1} R^2 \frac{2}{t_0}))}, \tag{S91}$$

where we used (S62). The proof is complete. $\qquad \square$

### S8.11 Proof of Proposition S11

*Proof.* Similar as in the proof of Proposition S7, we can compute that

$$J_{h_i}(w) = I - \frac{\eta}{b} H^{-1} \sum_{j \in S_i} \nabla^2 \ell(w, z_j) = (1 - \eta\lambda H^{-1})I - \frac{\eta}{b} H^{-1} \sum_{j \in S_i} y_j^2 \ell''_\sigma \left(y_j a_j^T w\right) a_j a_j^T,$$

where $\{S_i\}_{i=1}^{m_b}$ is a partition of $\{1, 2, \ldots, n\}$ with $|S_i| = b$, where $i = 1, 2, \ldots, m_b$ with $m_b = n/b$, and $H$ is a positive-definite matrix with $0 \prec mI \preceq H \preceq MI$, and

$$\ell''_\sigma(z) = \frac{1}{\sigma} \frac{e^{-(1-z)/\sigma}}{(1 + e^{-(1-z)/\sigma})^2} \geq 0, \quad \|\ell''_\sigma\|_\infty = \ell''_\sigma(1) = \frac{1}{4\rho}.$$

Therefore, if $\eta \in (0, \frac{m}{\lambda + \|R\|^2/(4\rho)})$ where $R := \max_i \|a_i\|$, then

$$1 - \eta m^{-1}\lambda - \eta m^{-1} \frac{1}{4\rho} R^2 \leq \|J_{h_i}(w)\| \leq 1 - \eta M^{-1}\lambda.$$

This implies that

$$\overline{\dim}_H \mu_{W|\mathbf{S}_n} \leq \frac{\log m_b}{\log(1/(1 - \eta M^{-1}\lambda))} = \frac{\log(n/b)}{\log(1/(1 - \eta M^{-1}\lambda))}, \tag{S92}$$

where we use (S62). The proof is complete. $\qquad \square$

### S8.12 Proof of Proposition S12

*Proof.* By following the similar derivations as in Proposition 5, we obtain

$$J_{h_i}(w) = \left(1 - \eta\lambda H^{-1}\right)I - \frac{\eta}{b} H^{-1} \sum_{j \in S_i} \text{diag}\left(\left\{B_r^{(j)}\right\}_{r=1}^m\right), \tag{S93}$$

where $\{S_i\}_{i=1}^{m_b}$ is a partition of $\{1, 2, \ldots, n\}$ with $|S_i| = b$, where $i = 1, 2, \ldots, m_b$, with $m_b = n/b$, and $H$ is a positive-definite matrix and $0 \prec mI \preceq H \preceq MI$ for some $m, M > 0$, and $\text{diag}(\{B_r^{(i)}\}_{r=1}^m)$ denotes a block diagonal matrix with the matrices $B_r^{(i)}$ on the diagonal defined in Proposition 5. As in Proposition 5, there exists a constant $M_y > 0$ such that $\max_i \|y_i - \hat{y}_i\| \leq M_y$. Then for $\eta \in (0, \frac{m}{C+\lambda})$ and $\lambda > \frac{M}{m} C$ where $C := M_y \|b\|_\infty \|\sigma''\| R^2 + (\max_j \|v_j\|_\infty)^2$, we get

$$1 - \eta m^{-1}(C + \lambda) \leq \|J_{h_i}(w)\| \leq 1 - \eta \left(M^{-1}\lambda - m^{-1}C\right).$$

This implies that

$$\overline{\dim}_H \mu_{W|\mathbf{S}_n} \leq \frac{\log m_b}{\log(1/(1 - \eta(M^{-1}\lambda - m^{-1}C)))} = \frac{\log(n/b)}{\log(1/(1 - \eta(M^{-1}\lambda - m^{-1}C)))}, \tag{S94}$$

where we used (S62). The proof is complete. $\qquad \square$