# OpenReview forum: "Fractal Structure and Generalization Properties of Stochastic Optimization Algorithms"
_NeurIPS.cc/2021/Conference — NeurIPS 2021 Spotlight_

### Official Review · Reviewer_iGxg · 2021-07-08

**Rating:** 7
**Confidence:** 3

**Summary:**

The paper gives novel generalization bounds for models trained with SGD (or variants of SGD) based on the Hausdorff dimension. The main contribution is Theorem 1 which shows, under certain assumptions, that for a general class of models, their generalization bound after training with SGD can be bounded by a term that depends on the Hausdorff dimension of the stationary distribution which the stochastic process converges to. Next, several interpretations of this bound are given for simple models like least squares, and 2-layer NN. Finally, an experiment shows on several datasets that there is a correlation between the generalization capacity of NN and a certain term in the given bound.

**Limitations And Societal Impact:**

The authors adequately addressed the limitations and potential negative societal impact of their work.

**Main Review:**

This paper provides a novel view on generalization of models trained by stochastic algorithms, specifically SGD and its variants. This perspective is very interesting and I think this is a good contribution. In Section 4 the authors give more concrete examples of how these bounds are translated for standard models such as regularized linear models with square loss and two-layer neural networks (where only the first layer is trained). Finally, the experiments in Section 5 give a good motivation for why this generalization bound is tighter than known bounds.

Classical generalization bounds on neural networks are exponential in the network’s depth, and the bounds presented here have the potential of being dependant on other parameters (e.g. a bound on the Jacobian’s norm) which may not be necessarily exponential in the depth. I think this makes the bound important for our understanding of how neural networks really generalize.

There are a couple of minor issues which I would be happy to see the author’s response:

1) In the least-squares and logistic regression bounds, there seems to be no dependence on the input dimension (assuming that the inputs are bounded). This is a bit weird, to have the same bound for all input dimensions, I think the authors should explain this in more detail.

2) Suppose that SGD converges to some minimum over the dataset (either local or global), wouldn’t the distribution \mu_W be constant? Since the gradient w.r.t, all the data points in the train set are zero? In what cases this distribution is non-constant, or am I missing something?

3) Under what conditions can we ensure that the distribution \mu_W converges for standard models such as linear predictors or neural networks? i.e. what are the conditions on the model and data such that Eq. (7) and (8) are satisfied? I think elaborating on this would give a better understanding of what is required from the data/model in order for the stochastic process to converge to some distribution.

4) Does \alpha play a role in Corollary 1? Does the corollary hold even when \alpha is extremely small?

To conclude, I think the contribution of this paper is interesting and novel. I recommend acceptance.

--------Post rebuttal comment--------

I read the author's response and decided to maintain my score.

**Time Spent Reviewing:**

7

---

> ### Author Response · Authors · 2021-08-10
> **Response Letter**
>
> We are grateful to the reviewer for the positive and encouraging feedback. We are happy to see that the reviewer found our submission to be important, novel, and interesting. Our responses are provided below.
>
> **1 - Dependence on the input dimension:** The input dimension potentially affects the term R, which forms the bound for the input data, and hence the input dimension will indirectly affect the generalization bound. Thanks for this comment. We will clarify this further.
>
> **2 - Convergence to a singleton:** If proper decaying stepsize is used within SGD (such as a stepsize that decays like ~ 1/k), the iterates can converge to a global or local minimum in which case the limit will be a singleton and the distribution will be concentrated at a point as you mentioned. However, when constant stepsize is used, as in the setting of our paper, due to the inherent/persistent noise in the SGD updates the iterates will oscillate and the limit will not be a point but a distribution. For example, in the toy example we illustrated in Figure 1 with quadratics in dimension one, the limit is a distribution whose support is the Cantor set. We will make this point clearer.
>
> **3 - Conditions to ensure convergence:** We thank the reviewer for the suggestion. We will elaborate this in the final version of our paper. For simplicity, we demonstrate how to validate this condition on the least squares problem
>
> $$\hat{R}(w) = \frac{1}{n}\sum_{i=1}^n (y_i - x_i^Tw)^2 $$
> where $x_i\in \mathbb{R}^d$’s are isotropic sub-Gaussian random vectors.
>
> Consider a uniform subset $S\subset \{1,...,n\}$ of size $b$. By the properties of sub-Gaussian random vectors (e.g. see [1, Thm 4.7.1]), we have $\mathbb{E}[\| I - \frac{1}{b}\sum_{i\in S}x_ix_i^T\|] \leq CK^2\sqrt{\frac{d}{b}}$ where $C$ is a positive constant, $K$ is the sub-Gaussian norm of $x_i$, which is true for a sufficiently large batch size.
>
> Now, consider the SGD update with mini-batch size $b$
>
> $$w_{k+1} = w_k - \eta \frac{1}{b}\sum_{i\in S} x_i(x_i^T w - y_i)  =: h_S (w_k, S_n).$$
>
> We look at the Lipschitz constant of this function system: For any $S$,
>
> $$\|h_S (w, S_n) - h_S (w’, S_n)\| = \|w- w’ - \eta \frac{1}{b}\sum_{i\in S} x_ix_i^T (w - w’)  \| \leq \|I - \eta \frac{1}{b}\sum_{i\in S} x_ix_i^T \| \|w-w’\|$$
>
> Notice that the quantity $\|I - \eta \frac{1}{b}\sum_{i\in S} x_ix_i^T \|$ is an upper bound on the Lipschitz constant $L_S$ of $h_S$. Investigating the condition (8), we have $\mathbb{E}[\log(L_S)]\leq \log\left(\eta CK^2 \sqrt{\frac{d}{b}} + (1-\eta)\right)$, where the first step follows from Jensen’s inequality and the second follows from sub-Gaussian property. The right hand size of the above bound can be made smaller than 0 for a sufficiently small step size choice $\eta$.
>
> [1] High-Dimensional probability: An Introduction with Applications in Data Science, Roman Vershynin, 2020.
>
> **4 - The role of $\alpha$:** The Holder condition is mainly used to ensure that the invariant measure exists and the constant $\alpha$ does not directly interact with the bound. However, it might affect the rate of convergence to the invariant measure. We will clarify this point in the revised version.

---

### Official Review · Reviewer_PfjU · 2021-07-16

**Rating:** 7
**Confidence:** 4

**Summary:**

{\bf Summary:} The authors study generalization bounds for stochastic gradient algorithms by studying the invariant distribution of the algorithm. The bounds involve the Hausdorff dimension of the invariant measure, which the authors proceed to bound in several applications in terms of hyperparameters of the algorithm (step-size, batch-size, batch-selection mechanism), parameters of the application (Hessian of risk function, data distribution) and an {\em integral} quantity of the invariant distribution, which can be estimated more easily.


**Limitations And Societal Impact:**

the authors have adequately addressed the limitations and potential negative societal impact of their work.

**Main Review:**

{\bf General assessment:} The article is well-written and the topic is interesting. It bridges topics in fractal geometry and stochastic optimization and generally does both fields justice. While it is on the theoretical side, the results are interesting enough that I believe they merit inclusion in NeurIPS21. I am not quite familiar enough with the surrounding literature to assess the novelty of the results perfectly, but my impression is that the progress made in this paper is sufficient to be included.

My biggest concern is that the authors obtain an error bound which cannot be evaluated and is as such not useful in practice at this point. The invariant which the authors study is positively correlated with generalization behavior, but only explains the first order correction to an error bound, not the leading order term. Furthermore, the invariant measure is only relevant asymptotically, and the authors do not obtain rates (which they claim as a subject of future work).

Nevertheless, I believe that the analysis is quite interesting and sheds light on the properties of stochastic optimization algorithms, which may open the door for further advances.


{\bf Specific comments:}
There are a few aspects of the article that should be improved before publication, in particular in Corollary 1 and Proposition 5. I am not entirely convinced by the numerical results, but might be missing something. If the authors can make such improvements, I would be in favor of publication.

* l 147: The supremum should be taken for $s\geq 0$ to allow Hausdorff dimension zero without having to discuss the supremum of the empty set.

* l 180: SGD or Newton's method have no 'memory' and are Markov processes, as is clear from the IFS representation formula. SGD with momentum, ADAM etc.\ have a memory of previous states, unless they are viewed on a higher-dimensional phase space. Could the authors briefly comment on how their results extend to this case, since even the appendix avoids this case.
* l 185: I am not sure this is a more realistic set-up -- typically, we shuffle the data set between epochs, so that the probability of getting the exact same batch twice is pretty low, and the probability of seeing the same batch in consecutive time-steps would be zero in practice (except between epochs).
* l 187: I believe the authors are introducing the new notation $p_i$ here, not making a condition on whether or not the iteration step can be re-written in a certain way.
* l 190: Again, this is not the realistic setting, since the probability of seeing a batch would depend on whether this batch has  already been used during the epoch. Uniform sampling is not the default option in practice, since we do not repeat the same batch twice in a row. In Corollary 1, the implication is that $p_i$ is constant in time.
* l 239: Is $n$ needed in the definition of $\mathfrak G$?
* l 240: I assume that $\mathbb P = \mu_{W|S_n}$ here and supposed to hold uniformly over $S$ and $n\in\mathbb N $, at least almost surely? This is not clear at first reading, please explain.
* l 241: This condition looks like a quantitative bound on the independence of the sigma algebras $\mathfrak F, \mathfrak G$. I understand that space is limited and that references are given, but a few more words on the intuition would be appreciated.
* l 251: Any quantitative estimate on how large $n$ has to be?
* l 258: The decay of $\log(\log n) / \log n$ cannot be considered as 'very rapid'. The condition is rather similar to a lower bound on the upper Hausdorff dimension of $\mu_{W|S_n}$. Do the authors have an idea under which conditions, $\overline{dim}_H (\mu|_{W|S_n})\to 0$? This seems to be the case in e.g. strongly convex problems or problems with a PL property and a unique minimizer (with small enough learning rate and noise which vanishes at the minimum), but unrealistic in the context of deep learning, where the set of minimizers is high-dimensional.
* l 262: It is unclear how $\alpha$ and the Hölder constant enter in the result. I suspect the constant $M$, but if this could be mentioned (or even quantified), I would be grateful. This seems to be important in the discussion in l 305 as well.
* l 263: $h_i' = J_{h_i}$, I believe. (Also Theorem S.2 in the appendix)
* eq (11): Ent looks like an operator, not a variable. The upright typesetting of a three letter name is confusing here and should be changed.
* eq (11): Does the condition (8) guarantee that the first term under the square root is positive? Otherwise, the expression under the square root is negative for large $n$ (once $\log n$ is large enough). Can the authors comment on this more explicitly, and whether (8) has geometric implications for the objective functions in an SGD algorithm.
* eq (11): It seems that if $h_i$ is almost constant or very rough (so that $\log\|J_{h_i}\|$ is very large), then the first term drops out of the bound. I assume this is compensated for by the constant $M$? Even with a global Hölder bound, I do not see why the first term couldn't be made arbitrarily small. Is this reflected in the geometry of the invariant measure somehow?
* l 266 (or notation section): The operator norm depends on the norm of the underlying spaces. I assume this is the Euclidean norm everywhere in the article, but this might be worth noting (also for eq (7) and condition H3).
* Section 4: All loss functionals seem to be considered with an additional regularizer. Could the authors discuss
\begin{itemize}
* How their estimates relate to the estimates that would be obtained directly from regularization for minimizers (or e.g.\ low loss functions)
* How the regularization is incorporated in the time-stepping algorithm (e.g.\ in every batch, or as its own 'batch' which may be chosen stochastically, ...)
\end{itemize}
* l 299: $\ell(w)$ is not defined, strictly speaking. It may be advisable to denote $\nabla_w \ell(w,z_i)$ to be particularly clear.
* l 302: The bound is decreasing in $\eta$, but
\[
\lim_{\eta\to 0^+} \frac{\log(n/b)} {\log\left(\frac1{1-\eta \lambda}\right)} = \frac{\log (n/b)}{\lim_{z\to 1^+} \log (z)} = +\infty.
\]
Also in the gradient flow limit, the algorithm becomes deterministic. Why do the authors expect this kind of behavior/a continuous invariant measure, rather than a single point and such different behavior compared to $n=b$? This seems to be in line with Figure 1, but not Figure 2. Is the scale the same in all plots in Figure 2? The 'continuous' behavior seems to still result from fairly large step size. Would the authors expect a double descent-like phenomenon?
* l 311: If $m_b$ is taken to be constant over large data sets, the benefits of SGD in terms of performance decrease. Since the dependence on $n/b$ is only logarithmic, it seems that a tradeoff with moderately large/moderately growing $m_b$ would be reasonable.
* l 318: better: 'that $\eta<1/\lambda$ and that the data...'
* l 319: Unlike logistic regression, this loss seems to force $y_i = \langle w, a_i\rangle$ instead of having the correct sign and allowing variability in magnitude between data points. This seems more in line with a least squares regression than logistic regression. Could the authors comment on why they dub this a 'nonconvex formulation of logistic regression' or give a reference?
* l 322: Should this be $|t|^2/t_0^2$ in $\rho_{exp}$?
* eq (17) and Proposition 4: Why was $\lambda$ changed to $\lambda_r$ comparing to previous notation?
* l 331: 'smooth' is a very imprecise term here. How smooth, and are quantitative global bounds required? It seems that $\sigma \in W^{2,\infty}$ or $\sigma'' \in L^\infty$ might be natural?
* Proposition 5: $v_i$ seems to be a matrix, which is very confusing notation. Commas between the columns would make the expression easier to read. What norm do the authors use here -- is this the element-wise $\ell^\infty$-norm or an operator norm? If $v_i$ is meant to be a (very long) vector, please be more explicit in the description.
* Proposition 5: This seems to be written up backwards, with multiple 'where ...' explanations following each other. Can this be rewritten in a more intuitive way? If necessary, the precise statement could be postponed to the supplementary material.
* Proposition 5: $\hat y_i$ is the output of the neural network. $v_i$ depends on the weights of the network. How are $M_y$ and $\|v_i\|$ controlled during the iteration/is it sufficient to consider bounds at initialization/are there a priori estimates for the invariant measure/...?
* Proposition 5: Since $\lambda$ has to be lower bounded, can the authors discuss the dependence of $C$ on $m$ and $n$? Can we e.g.\ choose small $\lambda$ for large data sets and wide networks?
* l 334: Which norm of $\sigma''$ is taken here?

* l 343: The claim that a numerical method is developed is very generous, as the authors essentially do the natural thing and do not provide error estimates. The non-trivial idea seems to be in how to find the operator norm of a large matrix, which is barely discussed in the article and the supplementary material. However, I would question whether 200 {\em consecutive} iterates of SGD can be considered as iid samples from the invariant distribution. This seems like a too short time window (or too little time between the individual iterates). I would encourage the authors to rerun the simulations or discuss why this approach is valid.
* l 345: $R$ was used in a different way above. Please stay consistent in notation and don't switch like this.
* l 345: The error bound depends on $\sqrt{R}$, which is only approximately linear if the second term in the error bound is fairly large compared to $R$, which seems to be true in all except the first plot (the total generalization error is much large than the variability within the graph). I assume that this is why the authors chose to consider $R$ instead of $\sqrt R$.

However, this may suggest that the authors closely investigate a term which is not the main contributor to the generalization error, much like estimating the first order term of a Taylor series, but not the term of zeroth order. Could the authors explain why their results are informative for consideration in practice?

Using a Taylor approximation of the square root $\sqrt{Q+\eps} \approx \sqrt Q + \frac1{2\sqrt{Q}}\eps$, can the explicit error bound be fit quantitatively with reasonable accuracy, where $\eps$ is the dimension of the invariant measure (centered at average) and $Q$ is the generalization error in the middle of the domain?
* l 362: The authors claim the bound of Corollary 1 to be informative, but argue in l 338 that the bound cannot be evaluated, which makes it not informative. They furthermore design their method to only evaluate part of the bound, which contributes less to the overall error. While I find the bound interesting, I do not believe it to be useful in practice for this very reason (without further improvements), and I find the formulation here misleading.
* l 363: In convex models where a unique minimizer exists, the Hausdorff dimension of the invariant measure should be zero if the time-step size is below $2/L$, where $L$ is the Lipschitz-constant of the gradient of the objective function. In unregularized overparametrized models, there should be a manifold of dimension $m-n$ on which the empirical loss vanishes exactly, and any measure which is supported on this manifold is invariant under SGD dynamics since the data labels are matched exactly. As outlined above, I do not quite see why the complexity should decrease with $\eta$ (at least below a certain threshold), and I believe a more thorough discussion is required here, which might not be possible within the framework of this article. Is there some intuition the authors can provide for the continuous time limit or small step sizes? The simplified picture presented here does not seem to do the complexity of the situation justice, as the authors acknowledge.




* l 599: The authors did not introduce SOSC or ROSC. Does the result involve $\alpha$ in any quantitative way?
* l 622: The authors suggest here that $H$ is fixed, but consider $H$ related to the Jacobian below. I am not sure which Jacobian is considered here: The gradient of the objective function (which would make $H$ rank one) or its Hessian (the Jacobian of the gradient). In either case, the ideas are only compatible for quadratic objective functions.
* Algorithm 1: Gradient of $f$? and $g_\theta = g_w$.


* l 729: The proof seems to be correct, as far as I can tell, but it is challenging to read. It would be helpful if it could be broken up into a series of steps, and/or if the authors could provide an outline and heuristic before the technical details.

My understanding is that the key argument is a concentration inequality/tail bound, and that the Hausdorff dimension appears due to a logarithm of the covering numbers. The major technical complications seem to stem from the randomness in $S$. Am I missing something central with this interpretation?

* l 737: I believe that a countable subsequence in $\delta$ must be selected. This can potentially be avoided here due to monotonicity, but should be discussed.
* l 745: It would be helpful if the authors could say a little more about the distribution $\mathbb P$ and argue why the sets they consider are measurable under this distribution. As the choice of $A_{S_n,k}$ is highly non-unique over $S$, it would surprise me if there weren't selections for which the probabilities are not well-defined (but I do imagine that a measurable selection can be found -- Chapter 14 in {\em Infinite dimensional analysis} by Aliprantis and Border is probably a good place to look)
* eq (S39): Is this used below? EDIT: I found it in l 763, significantly later. Please refer back to (S39) for the definition or make the definition closer to where the set is needed.

* l 782: Norms missing in the equation above
* eqs (S59), (S60): I think the determination which term is larger can be made explicitly due to the choice of $\eta$. $\gamma_i>0$ since it is the minimum of positive terms... The condition $\eta < \frac{1}{R^2+\lambda}$ was already made.
* l 783: Do you need to divide by $\|w\|$ in the equation above?
* l 785: There is  $\binom{n}{b}$ instead $n/b$, which is used in (S63). This should be cleared up. The confusion seems to be a consequence of being able to select either {\em any} subset of size $b$ or only disjoint batches.
The estimates for the two types of batched SGD are very different. Is it reasonable to expect that they would be so different?
* l 812: The output is {\em only} bounded a priori since the final layer is fixed at initialization. This needs to be pointed out.
* l 814: Missing space.

I only skim read results in the supplementary material and not in the main text.


**Time Spent Reviewing:**

5

---

> ### Author Response · Authors · 2021-08-10
> **Response letter (Part 1)**
>
> We thank the reviewer for the very detailed, constructive, and positive feedback. We are happy to see that the reviewer found our submission well-written and interesting.
>
> We will implement all the minor comments that the reviewer suggested (e.g., typos, inconsistencies in the notation, wording). Our detailed response is as follows.
>
>  **L180:** Basically, as the reviewer correctly pointed out, when there is a momentum, e.g. Nesterov accelerated SGD, ADAM, stochastic heavy ball method, we can augment the state space so that the augmented iterations are Markovian (see e.g. Example 1 and Example 2 from [Hodgkinson and Mahoney, ICML 2021]). If the stepsize is constant, the iterates still form an IFS and we believe we can still obtain bounds on the Hausdorff dimensions of the invariant measures of the *extended variables*. However, we need to consider marginals of such invariant measures on the extended space to be able to derive generalization bounds, which is not straightforward. This will be left as a future project.
>
> **L185, L190:** We agree with the reviewer that non-uniform sampling strategies are more common in practice. However, to ease the analysis we have resorted to uniform sampling strategies to avoid obscuring the main messages of the paper. In this sense, we believe that dividing the dataset into small chunks and sampling among them would be “more realistic” than uniformly sampling data points at each iteration, if not the most realistic. On the other hand, we can easily extend our theory to *state-dependent* $p_i$, whereas allowing for time-dependent $p_i$ seems highly non-trivial. We will clarify this point in the text.
>
> **L239:**   $n$ is indeed needed in the definition, but there is a typo. Here the set $A$ should be replaced by the set $A_{S_n, \delta}$ defined in Proposition 1. The $\sigma$-algebra is then defined by all such random variables with $n\geq 1$ and delta positive rationals. We have given the complete correction in our response to Reviewer gFd8.
>
> **L240:** The probability measure here is $\pi^\infty$, the law of an infinite sample sequence defined in the beginning of Section 3. As mentioned above there is a typo in the definition of the sigma algebra $\mathfrak{G}$.
>
> **L241:** The assumption attempts to control the dependence of the *topological properties of the support* of the invariant measure on the empirical loss at w. For an algorithm with high-stochasticity, the “support” of the invariant measure should not depend too strongly on the loss. We don’t expect the assumption to hold in full generality however, especially for “deterministic” algorithms. We will include more discussion on this assumption.
>
> **L251:** The notions of Minkowski and Hausdorff dimensions are essentially asymptotic, which unfortunately prevents us from obtaining any truly non-asymptotic result.  However, obtaining nonasymptotic results might be possible with further assumptions on the fractal dimension of the invariant measures and their supports. We will add a remark on this point.
>
> **L258:** We note that if the Hausdorff dimension decays faster than $\log\log(n)/\log(n)$, then the generalization error simply decays with rate $\log\log(n)/n$ independent of Hausdorff dimension.
>
> Indeed, the Hausdorf dimension will converge to 0 as $n\to\infty$ if for example SGD produces a countable set of local minima. This would happen for example if the loss landscape is strongly convex (or satisfies PL inequality) **locally** around each minima and one uses a decreasing step-size schedule to ensure that SGD converges to a point asymptotically. However, as the reviewer pointed out, neither asymptotic convergence of SGD nor local strong convexity is realistic in the context of deep learning where small perturbations around local minima produce the same loss value.
>
> **L262:** The Holder condition is mainly used to ensure that the invariant measure exists and the constant $\alpha$ does not directly interact with the bound. However, it might affect the rate of convergence to the invariant measure. We will clarify this point in the revised version.
>
> **Eq11 - positivity:** Correct, the term under the square root is positive if condition (8) holds, please see the paper [Ram06]. As you mentioned, (8) has geometric implications: It tells that the loss surface is such that SGD iterations are “contractive on average” in the sense that if $L_B$ is the Lipschitz constant when batch $B$ is selected; $log (L_B)$ is negative in expectation.
>
> **Eq11 - small first term:** Indeed, the first term can be arbitrarily small but then the second term can be larger. This is discussed right after Proposition 2 where we discuss that as the batch size gets larger and becomes equal to the number of samples $n$, SGD iterations behave like GD (where hi is constant) and the invariant measure will become a singleton at a local minimum. In this case, the first term will get arbitrarily small but as you mentioned this will be compensated by the second term $M$ which can get arbitrarily large.
>
> **L266:** The norm of the underlying spaces is indeed the Euclidean norm. We will mention this in the revised paper.
>
> **Section4 - regularization** This is an interesting question. At this stage we are not able to relate our estimates to the ones that would be obtained via directly choosing the “most regular” minimizer. On the other hand, regularization is incorporated in every batch as we already include the regularizer into the loss function itself (see e.g. (13)).
>
> **L302:** We shall note that our theory is only applicable to discrete-time recursions and the limit when $\eta \to 0$ cannot be investigated with our approach and different tools need to be used, such as the ones in [SSDE20]. That being said, directly analyzing the discrete-time recursion is one of our contributions as the behavior of the continuous-time limit can be significantly different for moderately large step-sizes, which is the common practice. On the other hand, in the case where $n=b$ the algorithm becomes deterministic, which is a vastly different setting compared to the stochastic case (e.g., consider the technical differences in ODEs and SDEs). In such a setting the dimension of the invariant measure becomes 0 and the term $M$ is potentially infinite. In this sense, our framework does not cover all potential settings, but it arguably covers the current deep learning practice in terms of moderate $\eta$ and $b << n$. Finally, the scales in Figure 2 are the same.
>
> **L311:** We thank the reviewer for making this observation, which escaped our notice. We can indeed allow for moderately growing $m_b$. We will modify the corresponding part.
>
> **L319:** Thank you, this is a typo; here we meant “nonconvex formulation for robust regression” following the literature, e.g. [Mei et al. The Annals of Statistics 46 (6A), 2747-2774, 2018]. We will correct this typo.
>
> **L331:** Here, as you mentioned, it would suffice to have $\rho$ to be twice continuously differentiable with bounded second derivatives. We will clarify this point.
>
> **Proposition 5 - $v_i$:** $v_i$ is a vector. The norm on $v_i$ is the element wise $\ell_\infty$-norm. We will add commas between the columns in the expression of $v_i$.
>
> **Proposition 5 - How are My and |vi| controlled:** We assume that $M_y$ and $|v_i|$ are uniformly bounded for mathematical convenience. We will explore it in the future if we can relax this assumption.
>
> **Proposition 5 - dependence of C on m and n:** We assume that $C$ is uniformly bounded for mathematical convenience. However, in general, $C$ by its definition might increase in m and n. So we do not expect that we can choose small lambda for large data sets and wide networks. We will add a note about this matter.
>
> **L334:** We take the $\ell_\infty$ norm. We will fix this typo.
>
> **L343:** We note that the 200 consecutive iterates of SGD are taken after the neural networks have been trained to convergence -- as measured by the incremental changes in the loss and accuracy the model achieves at each training step (see Appendix F for the full details of the convergence criteria). We note that this corresponds to over 24 hours of training for the larger VGG16 models. As such the 200 iterates, drawn subsequent to convergence, can then be reasonably considered to be drawn from the invariant distribution.
>
> The numerical method for the estimation of the operator norm for large matrices is presented in full in Appendix E.1. Though we admit that this method does not provide error estimates, it is essentially a subtle and non-obvious re-tooling of the power-iteration method of  [YGKM20] for the estimation of the eigenvalues of neural network Hessians (a very closely related object to the operator norm). Though this method does not provide error estimates, it is guaranteed to converge to the true operator norm if the top eigenvalue of the operator is ‘dominant’ (it is strictly larger in absolute value than other eigenvalues). Empirically we find that this condition holds true in our situation, and as such we can consider the operator norm estimates we provide to be close to their true value given a sufficient number of iterations of Algorithm 1. We will ensure to include our empirical results that support this.
>
> **L345:** It seems that there might be a misunderstanding and we apologize for the confusion. The reason why we plotted $R$ directly was solely because it appears as the most interesting term in the bound and we wanted to monitor its relationship with the generalization error. Hence, we simply omitted the square root. Similar considerations have been made in the literature, eg., [NBMS17, SSDE20]. We have now verified that very similar results are obtained when we use $\sqrt{R}$ instead of $R$, and we also obtain very high statistical significance. We will mention this explicitly in the text to avoid confusion.

---

> > ### Author Response · Authors · 2021-08-10
> > **Response letter (Part 2)**
> >
> > **L362:** The reason why we claimed that our bound is informative is that, experimentally, we observed that the first term (which is computable) correlates well with the generalization error and we believe that the second term can be neglected for a large $\eta/b$ ratio.  However, we understand the concern raised by the reviewer and we will rephrase the related sentences to make this point clear.
> >
> > **L363:** Indeed, for strongly convex minimization problems, if one uses constant step-size (**non-stochastic**) gradient descent with step $<1/L$, the algorithm will converge to a point, which has a 0 Hausdorff dimension. However, please note that we consider the **stochastic** setting: if one uses a constant step-size SGD, the algorithm is a Markov chain with a non-trivial invariant measure, and accordingly has a non-trivial Hausdorff dimension. For small enough step-sizes, this non-trivial invariant measure typically has the dimension $d$ (ambient dimension) and the dimension decreases as the recursion becomes *more chaotic* with large step-sizes. However, as we mention in a different part of our response, our framework is not appropriate for analyzing the continuous-time limit and different tools need to be used.
> >
> > **L599:** SOSC and ROSC denote strong open set condition and regular open set condition. We will add their definitions in the revised paper. Theorem 2.1 from [Ram06] only assumes alpha-Holder continuity for some alpha and the upper bound does not depend on alpha.
> >
> > **L622:** For our preconditioned SGD, indeed, $H$ is fixed. The discussions below are just providing an illustrative example that if we choose $H=JJ^T$, where $J$ is the Jacobian, then the problem is known as the Gauss-Newton method for least squares. Our preconditioned SGD method with fixed $H$ works beyond the quadratics; see Propositions S9-S12.
> >
> > **L729:** The interpretation is correct. The main difficulty is the fact that the set over which we are taking the supremum is random and depends on S. Additional difficulties arise due to the fact that fractal dimension is an asymptotic constant. We therefore need to appeal to Egorov’s lemma to obtain some uniformity. We will add a heuristic explanation/sketch of proof before the proof to highlight the main steps in the proof.
> >
> > **L737:** We do indeed choose a sequence of deltas, countable by definition. We will clarify this further.
> >
> > **L745:** Indeed there are measurability issues stemming from the selection of sets $A_{S_n, k}$. We understand that with additional technical work some of these issues may be resolved by appealing to measurable selection results as the referee suggests. We do feel however that the paper was already quite technical and dense and that a lengthy detour on measurability questions would detract from the main message of the paper. Although we agree this is not optimal, we stated in line 172 that “To avoid technical complications, throughout the paper we will assume that all the encountered functions and sets are measurable.” Alternatively we could have opted to state Proposition 1 as an assumption, that is assume the existence of measurable sets satisfying the desiderata in the beginning of Theorem 1.
> >
> > **L783:** We do not. This equation says that the norm of the Hessian is bounded between $\gamma_i$ and $\Gamma_i$.
> >
> > **L785:** As you pointed out, our theory can provide bounds in two settings  either *any* subset of size $b$ or only disjoint batches. To obtain these results, we build on the existing theory of iteration function systems; in particular we build on the results of [Ram06] which result in tighter bounds in the second (disjoint batch) setting. It would definitely be an interesting research direction to obtain tighter bounds in the first setting which will be an interesting future direction; this would require building a new theory for IFS that can go beyond [Ram06].
> >
> > **L812:** Thank you for this comment. We will point this out and clarify this further.

---

### Official Review · Reviewer_5gBo · 2021-07-22

**Rating:** 6
**Confidence:** 3

**Summary:**

The authors study the generalization error of iterative stochastic optimization algorithms whose iterates form an ergodic Markov chain. Particular examples include SGD and stochastic Newton methods for a large class of loss functions and models. The generalization error of these algorithms is upper-bounded by the Hausdorff of the invariant measure of the associated Markov chain. Bounds on this dimension are then given for some common example models and loss functions. Statistical correlation between the generalization error of deep neural networks and a quantity in the derived bounds is established numerically, but is only proven for a single layer model.

**Limitations And Societal Impact:**

I believe there are no potential negative societal impacts of this work.

**Main Review:**

The paper is well written and relatively easy to follow, although the notation is a bit heavy for a paper of this length. The results are interesting and I appreciate the various examples given as they provide intuition for what the bounds looks like when actually implemented in practice. I would like to have seen at least one of the examples where an explicit bound is given implemented numerically so the theory can be verified. It would also give an idea of how tight the bounds are since no lower bounds are proven. A major component missing from this work, however, is a discussion about the ramification of these bounds for various model/algorithms/data distributions. The authors say that their bound is able to capture specifics of the model, the algorithm, and the problem geometry (eq 11) however they never discuss this information specifically. How do two different models compare under a fixed data, loss function, and algorithm? How do two different loss functions compare under fixed data, model, and algorithm? How do two different algorithms compare under fixed data, loss function, and model? Etc.? Furthermore what do these bounds suggest for improving current optimization algorithms or developing new ones? Do they suggest any ways of tuning hyperparamters for optimal generalization? I think, at least partial answers to some of these questions should be given in discussion. Otherwise the paper feels as if it simply does math for its own, which is fine, but is not necessarily to NeurIPS.

**Time Spent Reviewing:**

3

---

> ### Author Response · Authors · 2021-08-10
> **Response letter**
>
> We are thankful to the reviewer for the constructive and positive feedback.
>
> **Examples with explicit bounds:** As the reviewer suggested, we will include (in the supplementary document) new synthetic data experiments on the models reported in Section 4, where analytical estimates are readily available to further illustrate our bounds.
>
> **Ramification of the bounds for various model/algorithms/data distributions:** We thank the reviewer for this insightful comment. To clarify the context, in the past few years, it has been widely agreed that the generalization bounds should explicitly capture the effects of algorithm/hyperparameters/architecture/data; yet, it is still not clear *what aspects* of the algorithm/hyperparameters/architecture/data have an effect on generalization. In other words, it is not clear how they should appear in the bounds. In this respect, we believe that our findings (Eq 12) form a novel contribution to this question, which might open up new directions.
>
> In terms of experimental validation, we shall note that we already have certain comparisons:
> * “hyperparameters” of the SGD algorithm -- all plots in Figure 3
> * “models” under the same algorithm/data/loss function -- Figure 3 - 2nd plot.
>
> Here we observed that our bound is informative in terms of comparing two different models and hyperparameters. However, we agree with the reviewer in that more comparison in this direction will be beneficial for better illustration. In the next revision, we will conduct experiments where we compare different optimization algorithms/architectures/loss functions in a controlled manner.
>
> **Improving existing algorithms:** The most prominent direction for how our theory can be exploited to obtain improved algorithms is to include the term of Eq 12 as a regularizer to the optimization problem and optimize over the learning-rates and batch-sizes as well. This approach would be an alternative to the methods that aim to “decrease the intrinsic dimension” and would provide adaptive learning-rates and batch-sizes that simultaneously decrease our upper-bound and optimize the loss function. As the reviewer suggested, we will add a discussion on this matter.
>
> Finally, we agree with the reviewer that doing math on its own would be less appropriate for NeurIPS; however, we believe that our paper should not be considered in such a category. We shall underline that our primary aim is to decrease the gap between the current deep learning theory and practice by developing a new theoretical framework that identifies new quantities which might help us better understand the often surprising behaviors in deep learning. We will emphasize this point in the introduction.

---

### Official Review · Reviewer_gFd8 · 2021-08-01

**Rating:** 7
**Confidence:** 3

**Summary:**

The authors give a general generalization bound for stochastic optimization algorithms, depending on the Hausdorff dimension of the invariant measure. This bound takes into account the dataset and training algorithm, and draws on theory on fractal dimensions and random iterated function systems in dynamical systems. The authors link this bound to quantities that can be estimated from running the algorithm in question - namely an expected logarithm of norm of Jacobian. They specialize the theory for least squares (SGD & stochastic Newton), logistic regression, SVMs, and 1-hidden-layer neural networks. They present experiments showing that this quantity does in fact correlate with with generalization performance for fully-connected and convolutional neural networks trained on CIFAR10 and SVHN.

**Limitations And Societal Impact:**

Yes.

**Main Review:**

Generalization in modern machine learning (with overparameterized deep nets) remains a mystery, and this paper provides some new ideas to tackle the problem. The generalization bound given in the paper is applicable to a wide range of stochastic optimization algorithms, and this is illustrated with corollaries of the main theorem. Moreover, experiments suggest that the proxy quantity they defined does indeed correlation with generalization performance in practice for neural networks. I am enthusiastic about the introduction of dynamical systems methods (which are not well-known in ML); this seems to be a promising technique to better understanding generalization.

The closest work is [ŞSDE20] which needs to assume that the trajectory is well-approximated by a SDE and instead control generalization error with the Hausdorff dimension of trajectories; this work instead considers the discrete-time process instead and bounds generalization error based on Hausdorff dimension of invariant measures, and potentially works in more general circumstances.

However, I am concerned about some technical issues.

(1) The definition of $\mathfrak G$ in H2 does not make sense to me. As defined, $1\{w\in N_\beta(A)\}$ is a deterministic value, not a random variable depending on $S\in \mathcal Z^\infty$. Also, why is $\{\alpha(S,n)\in J_m(\rho)\}$ in $\mathfrak G$?

Can the authors elaborate on the definition and dependence on M? Assumption H2 is technical and I don't understand what it means intuitively, or the connection to algorithmic stability. Why is this a reasonable assumption, and what kind of bound does one expect to be able to obtain for M? Importantly, why is M finite for reasonable algorithms? Intuitively, how would M vary as we change the amount of "stochasticity" (through step size, batch size) in the algorithm? As the authors explain, as the algorithm becomes deterministic, M can blow up, so wouldn't R (the complexity metric) be a poor metric for generalization in the regime of low noise?

(2) In the propositions, the Hausdorff dimension is bounded in terms of $\log(n/b)$. However, examining the proof, the bound is in terms of $\log(m_b)$ where $m_b=\binom nb$, so this gives a bound of $b\log n$. Such linear dependence on batch size would seem to be a serious shortcoming.


**Time Spent Reviewing:**

5

---

> ### Author Response · Authors · 2021-08-10
> **Rebuttal letter**
>
> We thank the reviewer for the time and careful examination of the paper and we are happy that the reviewer found the paper novel and widely applicable. We believe that we have addressed all the raised issues and hope that the reviewer could reconsider his/her score in the light of these explanations.
>
> **Definition of $\mathfrak{G}$:** Many thanks for spotting this typo. Here, the set $A$ should be essentially replaced by the set $A_{S_n, \delta}$ defined in Proposition 1. More precisely, the definition of $\mathfrak{G}$ should be the $\sigma$-algebra generated by the variables:
> $1( {w\in N_{\beta}(A_{S_n, \delta})}): \varepsilon, \delta, \beta\in \mathbb{Q}_{>0}, w\in N_\beta, n \geq 1,$
>
> such that
>
> $\mu_{W|S_n}(A_{S_n,\delta}) \geq 1-\delta, \dim_M A_{S_n,\delta} \leq \dim_H \mu_{W|S_n} + \varepsilon $
>
> Since the set depends on the sample sequence S, it is now random and therefore the indicator is a proper random variable.
>
> On the other hand, the reason why $\alpha(\mathbf{S},n)$ is $\mathfrak{G}$-measurable is as follows:
>
> Notice that for any $0<\delta\in \mathbb{Q}$,
> $|N_{\beta_n}(A_{S_n,k})| = \sum_{w\in N_{\beta_n},k} \mathbb{1} (w\in N_{\beta_n}(A_{S_n,k}) ),$
>
> so that $|N_{\beta_n}(A_{S_n,k})|$ is $\mathfrak{G}$-measurable as a sum of $\mathfrak{G}$-measurable variables. Then $\alpha(\mathbf{S},n)$ is also $\mathfrak{G}$-measurable as a countable supremum of $\mathfrak{G}$-measurable random variables.
>
> **Definition and dependence on $M$** : This assumption controls the dependence between the support A of the invariant measure of the algorithm “concentrates” and the loss landscape. Both of these objects depend on the sample in a non-trivial way. The assumption controls essentially how dependent the loss at a parameter value w with whether w belongs to this set and therefore offers quantitative control on how much the topological structure of A depends on the sample.
>
> $M$ can be related to Renyi-mutual information and a priori there is no reason to expect $M$ to be finite for general algorithms; intuitively, however, the more stochasticity an algorithm incorporates the more we expect the set A and the loss landscape to decouple. For example, for a purely random algorithm (which ignores the sample) the two objects will be independent. In the other extreme, where the algorithm is deterministic given the sample $M$ may fail to be finite. Since we are controlling the generalisation error on the support, which is itself a random set depending on the sample, to make progress some assumption like this is necessary. Yet, the reviewer is right: our bounds may become less informative for very low noise. We will clarify these points in the paper.
>
> **The $\log m_b$ term:** In lines 184 and 186 we use two different definitions for $m_b$, which might have created confusion. In all the propositions we use the version where $m_b=n/b$, hence the statements are correct. We will correct the typo in line 785 and use a clearer notation to avoid confusion.

---

> > ### Comment · Reviewer_gFd8 · 2021-08-19
> > **Question**
> >
> > Thanks for the response. Can you expand on why $\alpha(\mathbf S, n)$ is $\mathfrak G$-measurable? It's defined as the upper Hausdorff dimension of a measure, which involves an inf over sets of measure 1; how can this be written in terms of the $A_{\mathbf S_n,\delta}$?

---

> > > ### Author Response · Authors · 2021-08-20
> > > **Response to the question**
> > >
> > > We sincerely thank the reviewer for insisting on this point. We totally agree with the reviewer that there is an imprecision here, which somehow escaped our notice while writing our first response. Below we fully clarify this point.
> > >
> > > In our proof, what needs to be measurable with respect to $\mathfrak{G}$ is in fact **not** $\alpha(S,n)$, but $d_M(S,n,k)$ (defined in Equation S45). And, $d_M(S,n,k)$ is **indeed** $\mathfrak{G}$-measurable: in our initial response (see above), we in fact explained why $d_M(S,n,k)$ is $\mathfrak{G}$-measurable, rather than $\alpha(S,n)$.
> > >
> > > However, the imprecision was caused by the fact that, in the display equation after Line 759, we immediately used the inequality S38, which essentially replaced $d_M(S,n,k)$ with $\alpha(S,n)$. Instead, the equation after Line 759 should read as
> > >
> > > $$[\mathbf{S} \in \mathfrak{Z}^{*} ] \subset \cap_{n} [|N_{\beta}(A_{\mathbf{S}_{n}, k})| \leq (\frac{1}{\beta})^{d_M(\mathbf{S}, n,k)+\xi_n c'(\beta)}]$$
> > >
> > >
> > > The proof follows the same way until Line765 (we only need to make a very minor change in the definitions of $\tilde{\alpha}$ and $\varepsilon$ in Equations S50 and S51).
> > >
> > > At this stage, to invoke Assumption H2, we only need $d_M(S,n,k)$ to be $\mathfrak{G}$-measurable (and we do not need $\alpha(S,n)$ to be $\mathfrak{G}$-measurable).
> > >
> > > Finally, after the display equation after Line767, we invoke the inequality inequality S38. Then the rest of the proof is the same.
> > >
> > > To recap: We mistakenly invoked the inequality S38 earlier than it should be invoked. The proof should first use $d_M$ until we invoke Assumption H2, and then once the assumption is invoked, we use the inequality S38.
> > >
> > > We shall note that the proof strategy/statement of the theorem remain unchanged with this update.
> > >
> > > That being said, since this part is merely a technical detail and not essential at all to the statement of the theorem, to simplify the arguments, we propose to modify the definition of $\mathfrak{G}$, such that $\alpha(S,n)$ is $\mathfrak{G}$-measurable by definition.
> > >
> > >
> > > As our proof is already quite technical and dense, such a simplification might be more beneficial in terms of avoiding obscuring the main message of the paper.
> > >
> > >
> > > We hope our explanation clarifies this point. If needed, we can further provide the full technical explanations (which we omitted now to avoid clutter).

---

> > > > ### Comment · Reviewer_gFd8 · 2021-08-25
> > > > **Technical concerns addressed**
> > > >
> > > > Thank you for the explanation; this addresses my technical concerns. I lean towards keeping the current assumption to avoid strengthening the assumptions. I think this is a mathematically strong paper and am raising my score to 7.
> > > >
> > > > Here are a few additional suggestions: (no reply necessary)
> > > >
> > > > The phrasing of Proposition 1 is confusing. It doesn't seem like $\delta$ needs to be a function of $\epsilon$, and it can be rephrased as "for any $\epsilon,\delta>0$..."
> > > >
> > > > The proof of Theorem 1 is not easy to read - please add a proof sketch highlighting the main ideas.
> > > >
> > > > It would be nice to comment on in what sense the bounds are or are not impacted by overparameterization?

---

> > > > > ### Author Response · Authors · 2021-08-27
> > > > > **Thank you very much**
> > > > >
> > > > > We are very happy to see that our responses have addressed the technical concerns. We will happily make the suggested improvements in the next revision.

---

### Author Response · Authors · 2021-08-10
**General response**

We sincerely thank all the four reviewers for their time and effort invested in reviewing our paper. We've addressed the reviewers' comments as separate responses, given below.

---

### Decision · Program_Chairs · 2021-09-27

**Decision:**

Accept (Spotlight)

**Comment:**

This is a strong paper, in which the authors prove generalization bounds for iterative stochastic algorithms based on the Hausdorff dimension of the invariant measure. The bounds are specialized for several applications like least squares (SGD & stochastic Newton), logistic regression, SVMs, and 1-hidden-layer neural networks. Some supporting empirical verification is also included. The proofs use tools from dynamical systems which are not so common in this are and seem like a promising lens on generalization.